# Emergent patterns of collective cell migration under tubular confinement

Wang Xi[1,2], Surabhi Sonam[1,3,4], Thuan Beng Saw [1,5], Benoit Ladoux [1,4] & Chwee Teck Lim[1,2,3,5,6]

Collective epithelial behaviors are essential for the development of lumens in organs. However, conventional assays of planar systems fail to replicate cell cohorts of tubular structures that advance in concerted ways on out-of-plane curved and confined surfaces, such as ductal elongation in vivo. Here, we mimic such coordinated tissue migration by forming lumens of epithelial cell sheets inside microtubes of 1–10 cell lengths in diameter. We show that these cell tubes reproduce the physiological apical–basal polarity, and have actin alignment, cell orientation, tissue organization, and migration modes that depend on the extent of tubular confinement and/or curvature. In contrast to flat constraint, the cell sheets in a highly constricted smaller microtube demonstrate slow motion with periodic relaxation, but fast overall movement in large microtubes. Altogether, our findings provide insights into the emerging migratory modes for epithelial migration and growth under tubular confinement, which are reminiscent of the in vivo scenario.

[1] Mechanobiology Institute, National University of Singapore, 5A Engineering Drive 1, Singapore 117411, Singapore. [2] Centre for Advanced 2D Materials and Graphene Research Centre, National University of Singapore, 6 Science Drive 2, Singapore 117546, Singapore. [3] Department of Biomedical Engineering and Department of Mechanical Engineering, National University of Singapore, Singapore 117575, Singapore. [4] Institut Jacques Monod, Université Paris Diderot & CNRS UMR 7592, 75205 Paris cedex 13, France. [5] NUS Graduate School of Integrative Sciences and Engineering, National University of Singapore, Singapore 117456, Singapore. [6] Biomedical Institute for Global Health Research and Technology, National University of Singapore, #14-01, MD6, 14 Medical Drive, Singapore 117599, Singapore. Wang Xi, Surabhi Sonam and Thuan Beng Saw contributed equally to this work. Correspondence and requests for materials should be addressed to B.L. (email: benoit.ladoux@ijm.fr) or to C.T.L. (email: ctlim@nus.edu.sg)

Many human internal organs contain epithelial lumens such as cysts and tubules, which are composed of curved epithelial monolayers enclosing a central cavity. The organization and development of these various epithelial luminal architectures aid in the essential functioning of the organs and are essential in organogenesis[1]. One common form of morphogenetic process that promotes epithelial tubulogenesis is the collective migration of cell cohorts while maintaining epithelial integrity[2–5]. For example, in mammalian mammary morphogenesis, ductal elongation is accomplished by the movement of a group of interconnected cells at the ductal tip[6]. Similarly, coordinated migration of epithelial cells contributes to the positioning of the zebrafish pronephric nephron segment boundaries and to the convolution of the proximal tubule[4]. Importantly, anomalies in these epithelial motilities have consequences for a series of diseases such as cancers[6–8]. Thus, understanding the key cellular processes in collective cell migration can provide significant insights into epithelial morphogenesis as well as contribute toward disease therapies.

The movement of interconnected cells during tubule formation commonly happens in complex physiological environments consisting of a plethora of physical features such as confined spaces with out-of-plane curvatures[2,9,10]. External physical cues are known to have profound impacts on epithelial architectures and the dynamics of multicellular assemblies on planar surfaces as well as in confined environments[11–15]. Spatial constraint has been highlighted to induce epithelial migration modes that differ from unrestricted flat microenvironments[16,17]. For instance, epithelial cell monolayers show diffusion-like motion in rectangular microchannels[18] but undergo epithelial–mesenchymal transition (EMT) when exposed to scattering periodic micropillar restriction[19]. In addition, the degree and geometry of confinements pose another regulation on patterns of collective cell migration. While cell monolayers demonstrate caterpillar-like migratory motion in narrow rectangular strips[12], they exhibit coordinated rotating motion under circular boundary restrictions[20,21]. Furthermore, the importance of in-plane curvature cues in modulating the polarization[22], proliferation[23], wound healing processes[24], and organization[25] of expanding epithelial sheets has been confirmed recently.

It is also noteworthy that most of the prior studies investigating the role of physical cues on tissue migration have mainly employed two-dimensional (2D) flat cell culture systems, whereas morphogenetic movements[26] or tumor progression[27] are facing out-of-plane spatial constrictions and signals. Also, the 2D approaches mainly study planar epithelial sheets whose topography is fundamentally different from that of lumens. On the other hand, conventional in vitro approaches for epithelial lumen

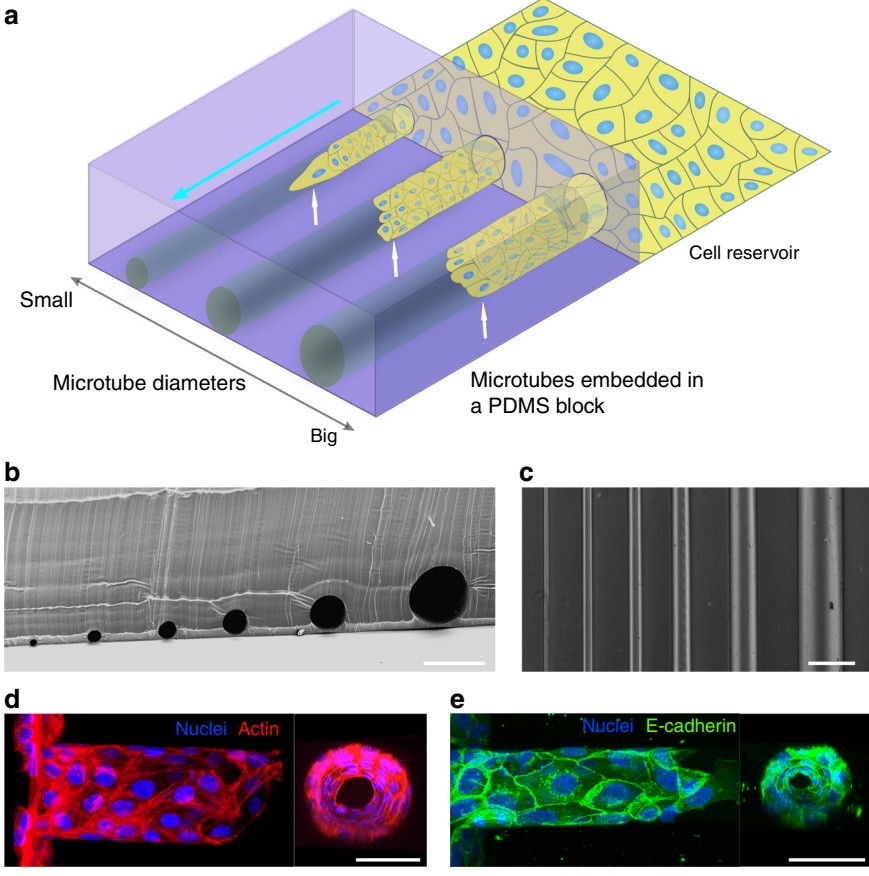

**Fig. 1** Formation of epithelial lumens inside PDMS microtubes. **a** Schematic illustration of the experimental setup. MDCK cells (blue, nuclei and yellow, cell body) are seeded on a fibronectin reservoir in front of a PDMS block containing cylindrical microtubes of different sizes. The cells start crawling into the openings of the microtubes once they are in full confluence (white arrows indicate the leading edge of the cohort and the cyan arrow, the direction of collective migration). **b**, **c** Representative SEM image (**b**) and optical image (**c**) of the microtubes embedded in the PDMS blocks (the diameters of the microtube: 25, 50, 75, 100, 150, and 250 μm from left to right). **d**, **e** Typical fluorescent z-stack projections showing groups of migrating MDCK cells inside microtubes: the cells were stained for nuclei (DAPI, blue) and in **d**, actin (phalloidin, red) or in **e**, E-cadherin (green). The cross-sectional views of the selected TCS sections in the right panels demonstrate the formation of hollow lumens inside the microtubes. Scale bars: 250 μm for **b** and **c**; 50 μm for **d** and **e**

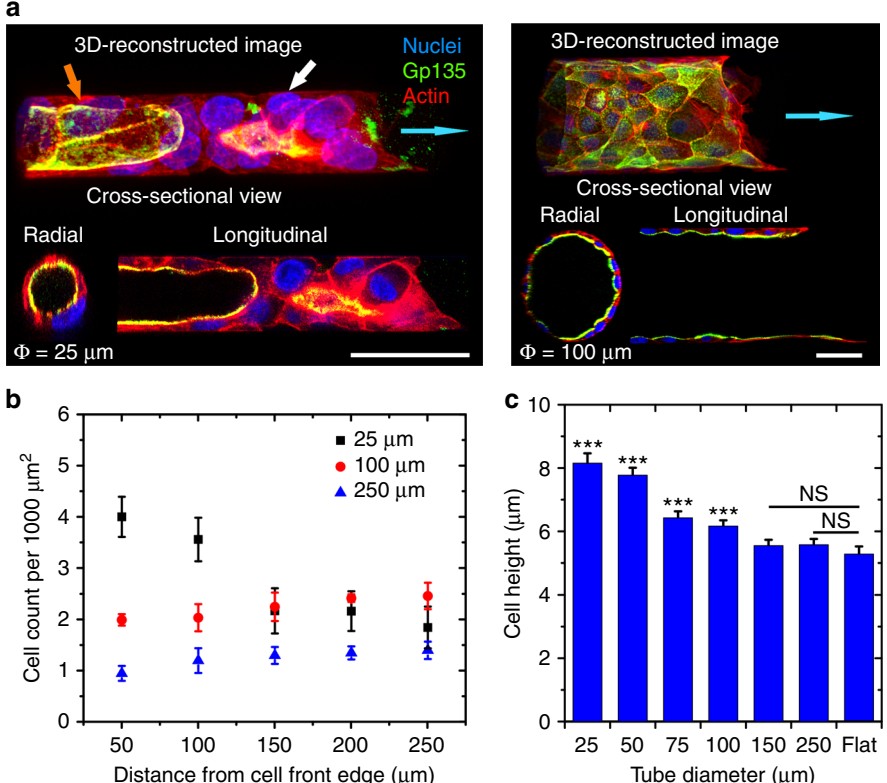

**Fig. 2** Epithelial TCS organization in microtubes of varying diameter. **a** 3D reconstructed fluorescent images of anti-Gp135- and phalloidin-stained MDCK cells (nuclei in blue, DAPI) in microtubes of different diameters (25 μm (left) and 100 μm (right)). Lower panels show radial and longitudinal cross-sections of the representative TCSs. Cyan arrows indicate the direction of the collective migration. White arrow indicates the blocked tube front and orange arrow, the formed epithelial lumen in a 25 μm microtube. Scale bars: 40 μm. **b** The cell density at different locations away from the TCS front edges in various microtubes ($n = 5$ from three independent experiments). **c** Average cell height of TCSs in different microtubes and flat surface ($n = 21$ from six independent experiments). $t$-test has been performed between each microtube diameter and flat condition, ***$P < 0.001$. NS non-significant. Data are presented as mean ± s.e.m.

formation involve making use of gels analogous to collagen matrices that encompass cells. Although such methods allow epithelial cells to reproduce tissue-like organization[28] and to mimic tubular branching morphogenesis in the presence of growth factors[28,29], the direction of epithelium advancement and lumen formation in gel-based systems is non-controllable, and thus renders the systematic study of epithelial dynamics in 3D environments very challenging. To this end, recent studies[25,30] grew cell sheets on the outer surfaces of cylindrical templates with varying diameter to investigate the collective cell behaviors in a more controllable manner. However, these systems resulted in epithelial tubules having inverted polarity that is incomparable with physiological situations[25] and no in-depth study on the dynamics was provided. While fabricating circular microchannels with conventional photolithography technique remains challenging, in some successful cases, cell monolayers that were cultured inside such channels under perfusion mainly investigated endothelialisation[31–33]. Up to now, most of the studies trying to reproduce epithelial cavity networks have aimed at understanding the molecular mechanisms responsible for lumen development, and very few have tried to unravel the dynamical aspects of coordinated epithelial behaviors across space and time.

To address these challenges, we provide a microtube platform for the study of collective epithelial dynamics that leads to the formation of hollow tubules. The microtubes have dimensions ranging from one to several cell lengths in dimension and serve as physical guiding cues for collective cell behaviors. We observe that epithelial Madin-Darby Canine Kidney (MDCK) and

MCF-10A cells demonstrated coordinated migration under these tubular confinements. Further, they form lumens enclosed by cell sheets and exhibit similar apicobasal polarity as seen in vivo, even for the tubes with the most restrictive conditions. Interestingly, the progression of these hollow, tubular epithelial cell sheets (TCSs) in microtubes smaller than three cell lengths (≤50 μm) are led by clusters of interconnected cells plugging the tubes, reminiscent of sprouting and branching morphogenesis[6]. We show that cells adopt different migratory patterns, which strongly depend on the extent of tubular constraint such as overall forward cell motion in large microtubes and onward and backward movements in small tubes. Our analysis reveal that these modes are distinct from that observed on flat tracks[12]. Additionally, the effect of tubular confinement on the epithelial architecture is reflected in actin stress fiber organization and cellular orientations. Altogether, our findings provide important clues to understand the pivotal role of external physical constraints in dictating epithelial morphogenesis.

## Results

**Formation of tubular epithelial cell sheet inside microtubes.** We used smooth platinum or copper wires with various diameters as templates to fabricate circular microchannels (microtubes) in polydimethylsiloxane (PDMS) to study tubular collective epithelial migration (Fig. 1a). Briefly, metal micro-wires were aligned in parallel just 1–2 μm above a silicon wafer using a precise stage (Supplementary Fig. 1). A PDMS polymer precursor

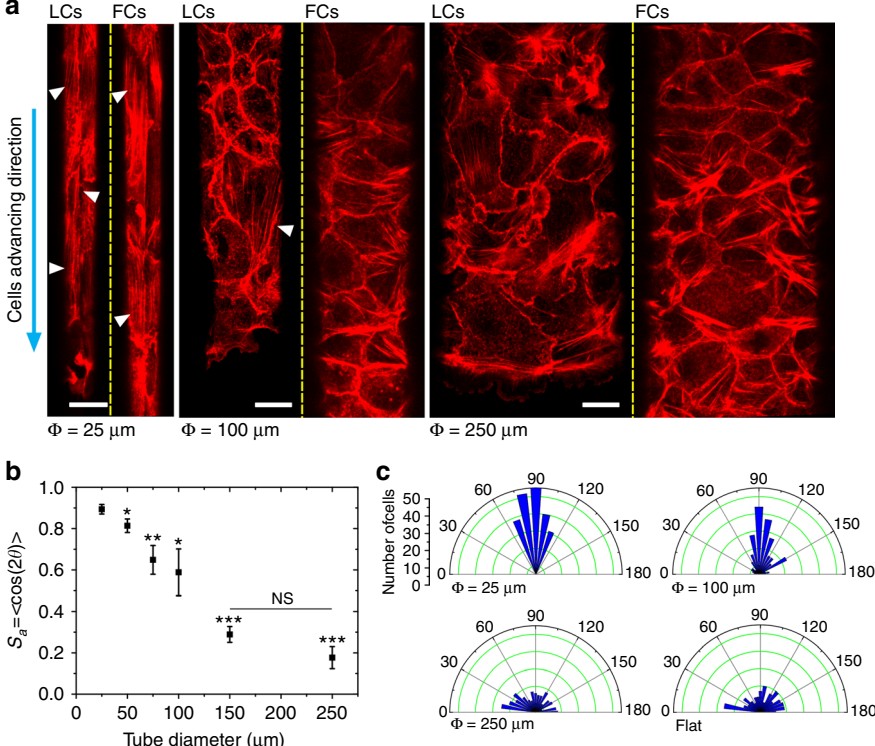

**Fig. 3** Epithelial cytoskeleton and cell orientation in different TCSs. **a** Phalloidin-stained MDCK TCSs exhibiting basal actin stress fiber organization in different tube dimensions (25 μm (left), 100 μm (middle), and 250 μm (right)). LC is the representative image of the cells at leading cell front of the tube and FC is the representative image of the following cells. Cyan arrow points in the direction of migration. White arrowheads point to the actin microfilaments oriented along the tube long axis. Scale bars: 15 μm. **b** Order parameter ($S_a = <\cos(2\theta)>$) of the actin filament as a function of microtube dimension, where $\theta$ is the angle of actin filament with respect to the tube length. 1 signifies parallel alignment and 0 signifies randomness in organization ($n = 10$ from five independent experiments). $t$-test has been performed between each microtube diameter and 25 μm, *$P < 0.05$, **$P < 0.01$, ***$P < 0.001$, NS non-significant. Data are presented as mean ± s.e.m. **c** Polar graph plotting the histogram of the cell orientation distribution inside microtubes (25, 100, and 250 μm (from left to right)). 90° means oriented along the microtube long axis, and 0° and 180° mean the cell is oriented perpendicular to the long axis

was then used to stencil the shapes of the metal wires. After cross-linking, the metal wires were removed from the polymerized PDMS, resulting in PDMS blocks embedded with straight circular microtubes (Fig. 1b, c; Supplementary Fig. 2) of exact diameters of the metal wires. This technique can produce transparent and biocompatible microtubes with smooth inner surfaces for long-term cell culture[34]. We used the microtubes with diameters of 25–250 μm in our experiments because 25 μm represents the diameter of a distal tubule (~30 μm) in kidney nephrons[35], while 250 μm is the approximate size of papillary collecting ducts (200–300 μm)[36]. The inner surfaces of the microtubes were functionalized with fibronectin before placing the PDMS block in front of a cohort of advancing MDCK cells. The narrow gap between the bottom of the microtubes and the substrate enabled MDCKs to crawl into the microtubes (Fig. 1a, d). The microtubes offer out-of-surface negative curvatures with tubular confinement, which is fundamentally different from the previous experimental setups based on cells cultured on planar surfaces[12,18,37] or borderless cylindrical confinement that provides positive curvatures[25]. We observed that in all circumstances, once cells wrapped around the whole circumference, they collectively migrated along the microtubes with typical lengths of ~1 mm while maintaining their cell–cell adhesion to form a hollow TCS (Fig. 1e; Supplementary Fig. 3; Supplementary Movie 1). Sur-prisingly, lumens were able to develop even in the smallest tubes with a diameter of only one or two cell lengths (~20–50 μm) (Fig. 1e). The epithelial lumens formed in our microtube platform

are reminiscent of various epithelial ducts with a broad range of diameters, including those formed physiologically[35,36] as well as those developed in 3D hydrogel[10].

**Tubular confinement influences epithelial TCS organization.** Epithelia probe external physical signals particularly through cell–substrate adhesions and the associated actin cytoskeleton, and thus adjust their organization and distribution according to the geometrical constraints[12,25,37,38]. In vivo extracellular matri-ces allow lumen formation of epithelial cells during which they undergo apicobasal polarization[39]. Thus, we first sought to determine the TCS polarization by immunostaining the cells with Gp135—an apical marker. The TCSs were fixed and imaged after the front had progressed inside the microtubes for 24–48 h to ensure the formation of lumens. The fluorescent 3D reconstruc-tion confirmed that MDCK TCSs polarized apicobasally in the tubes of all sizes with apical surfaces facing the lumens (Fig. 2a; Supplementary Fig. 4), recapitulating the polarized epithelial ducts in physiological conditions[10]. Notably, the cells at the tip of the smallest tubes (25–50 μm) lacked such polarity and instead of forming hollow TCSs, usually organized as multilayered epithelial structures (Fig. 2a; Supplementary Figs. 4, 5; Supplementary Movie 1), which plugged the microtube (Supplementary Fig. 6a). An analogous organization of multilayered epithelium at the ductal tip was also observed in both normal and neoplastic epithelia during morphogenesis[6]. Interestingly, after advancing

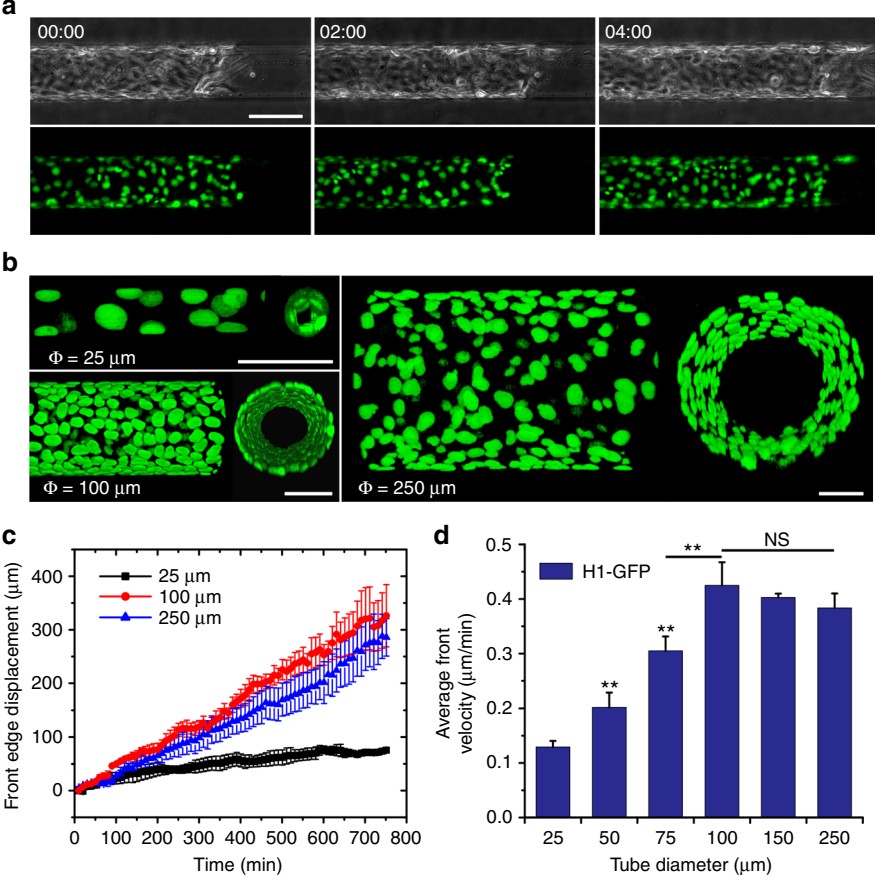

**Fig. 4** Collective migratory speed of MDCK TCSs varies with microtube diameter. **a** Representative time-lapse montage showing the collective migration of H1-GFP MDCK cells inside a 100 μm microtube. (top: phase contrast; bottom: H1-GFP showing the nuclei). Scale bar: 100 μm. **b** 3D fluorescent reconstruction of H1-GFP MDCK TCSs in different microtubes: top (left panel) and cross-sectional (right panel) views of the TCSs in each subfigure. Scale bars: 50 μm. **c** Displacement of MDCK cell fronts in different microtubes (25, 100, and 250 μm) as a function of time ($n = 3$ from three independent experiments). **d** Average velocity of cell front of H1-GFP MDCKs in tubes of different diameters (25, 50, 75, 100, 150, and 250 μm, from left to right), ($n = 6$–8 from 3–4 independent experiments in each condition). For each condition, $t$-test has been performed between each microtube diameter and 25 μm, unless otherwise indicated by line, $**P < 0.01$, NS non-significant. The plots represent the mean ± s.e.m.

deeper into these small microtubes, hollow lumens enclosed by epithelial monolayers of single to multiple cells formed at the back of the clusters of leader cells (Fig. 2a; Supplementary Fig. 4). In addition, in such small tubes, the cell density decreased from roughly 3–4 cells/1000 μm² at the leading edge to 1–2 cells/ 1000 μm² about 250 μm away from the front (Fig. 2b; Supplementary Fig. 6b). The abrupt decrease in cell density at *ca.* 50–100 μm from the leading front is due to the change from the multilayered cluster to hollow lumen (Fig. 2a, b; Supplementary Figs. 4, 6b). However, the slope of descending trend in cell count flattened in 75 μm microtube and was reversed in 100 μm and larger microtubes. The low cell density and increasing trends were also reported in epithelia cultured on strips of different 2D constraints[12], and this indicates that the epithelial organization in microtubes ≥100 μm approached that in planar tracks. More importantly, the transition in our experiment demonstrates the effect of tubular confinement and curvature on epithelial organization, which starts to disappear as the tubular diameter exceeds 50 μm. Furthermore, the average cell height in the TCSs was found to be inversely proportional to the tubular diameter (Fig. 2c), where the averaged cell height in the largest microtubes (150–250 μm) was similar to that measured on the flat substrates and the cells in the smallest microtubes were ~45% taller. Cells are known to spread less on bowl-shaped structures with negative

curvature in all directions[40,41] as is also observed in MDCK cells (Supplementary Fig. 6c). As the smallest microtubes have strong negative curvature in the circumferential direction, this could suggest that the cells spread less in these tubes and thus acquire a taller cell height. There is also a possibility that a significant radial force exists pointing toward the center axis of the microtube that can stretch cell toward the center of the tube (Supplementary Fig. 6d). Assuming that acto myosin contractility generates a tension, $T$, in the circumferential direction of the tube through cell–cell junctions, the resulting force per unit length exerted on the epithelial cells surrounding the tube is $T\rho\vec{n}$, where $\rho$ is the curvature and $\vec{n}$ is the normal vector to the border directed toward the interior of the tube. This radial force originated from tissue tension is proportional to curvature and may thus explain why cell height could be taller in smaller microtubes with higher curvature. Taken together, these findings identified a critical negative curvature of ~1/12.5–1/50/μm as the threshold for the MDCKs to respond to tubular constraint in a significant manner.

**Epithelial organization in microtubes.** The cellular cytoskeleton in terms of contractility and dynamical properties has been shown to respond to the external mechanical cues for single cells[42–44] and collectives[12,15,45]. Therefore, we studied how the changes in tubular confinement and curvature influenced the

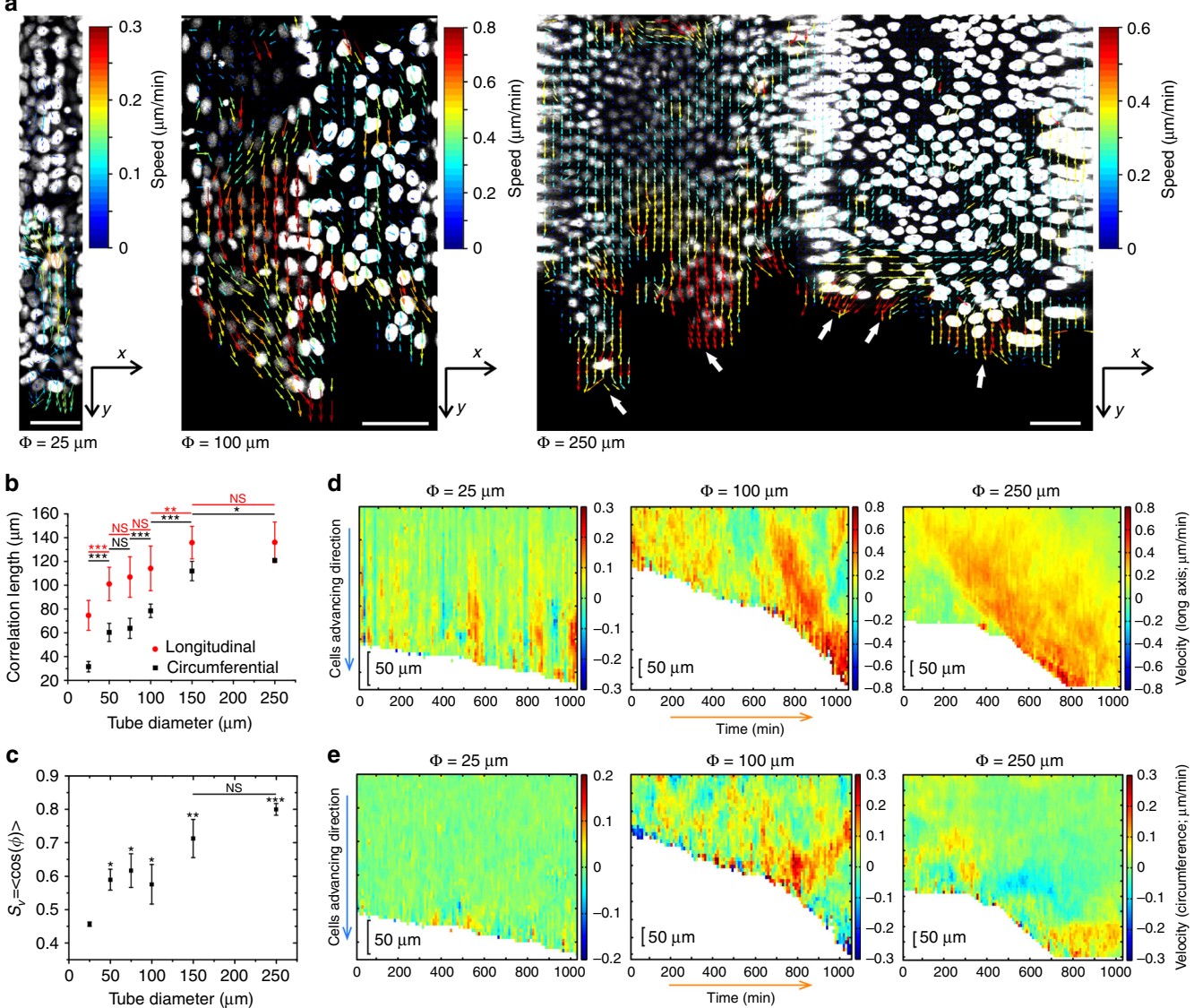

**Fig. 5** PIV analysis of MDCK cell tubes in microtubes of different diameters. **a** Direction of velocity fields showing backward and forward motion in 25 μm tubes (left), highly directed forward motion in 100 μm tubes (middle) and multiple leading edges (white arrows) in 250 μm tubes (right). Scale bars: 75 μm. **b** Longitudinal and circumferential correlation length of velocity vectors ($n = 14$ from five independent experiments). **c** Order parameter ($S_v = <\cos(\phi)>$) of the velocity vectors in MDCK TCSs ($n = 6$ from three independent experiments). **d** Kymograph of the average velocity parallel to the longitudinal axis of different microtube ($y$-direction in **a**, from left to right: 25, 100, and 250 μm). **e** Kymograph of the average velocity along the tubular circumferential of different microtube ($x$-direction in **a**, from left to right: 25, 100, and 250 μm). The plots represent the mean ± s.e.m. For each condition, $t$-test has been performed between each microtube diameter and 25 μm, unless otherwise indicated by lines, *$P < 0.05$, **$P < 0.01$, ***$P < 0.001$, NS non-significant

epithelial tissue organization by looking at high-resolution images of the cells' basal actin stress fibers (Fig. 3a). Parallel and aligned actin stress fibers were observed in the leading cell fronts (LCs) in TCS of all diameters up to 75–100 μm (Fig. 3a; Supplementary Fig. 7a). Interestingly, in TCS smaller than 75 μm, actin stress fibers were aligned longitudinally throughout the entire lumen (Fig. 3a; Supplementary Fig. 7a), while such alignment disappears in the follower cells of TCS in diameters ≥75 μm. To better quantify these findings, we computed the overall nematic actin order parameter defined as $S_a = <\cos(2\theta)> 25$ (Methods), which indeed decreased by almost five-fold for tube diameters ranging from 25 to 250 μm (Fig. 3b). In larger microtubes (≥100 μm), instead of forming well-aligned actin microfilaments, an inter-crossing actin cytoskeletal arrangement was observed, comparable

to the random orientation in MDCKs cultured on flat surfaces. We further wanted to understand whether the basal stress fiber orientation had any correlation with the orientation of the cell body. To this end, we plotted the cell body orientation defined as the angle formed between the major axis of the cell and the microtube longitudinal axis (Methods) as a function of the microtube diameter. As shown in the polar plot in Fig. 3c and Supplementary Fig. 7b, more cells aligned closer to the tube long axis for microtubes with diameters smaller than 100 μm. In contrast and as expected, a global isotropic cell orientation was observed in larger microtubes similar to that seen on flat surfaces (Fig. 3c). Altogether, these data demonstrate a clear correlation between cell shape and actin cytoskeleton orientations. The alterations in cell orientation and actin organization appeared at a

threshold of ~75–100 μm in diameter, consistent with stress fiber orientation.

**Tubular confinement induces different migration speeds.** Actin cytoskeletal organization is known to correlate with the migration dynamics of cell cohorts[46,47]. Thus, upon understanding the specific cell orientation and cytoskeletal organization in the TCS, we deemed it important to gauge the collective dynamics of the tissue migrating inside the microtubes. In order to study the collective migration patterns in TCSs, we cultured MDCKs that stably express Histone1-GFP (H1-GFP) in various sized microtubes (Fig. 4a; Supplementary Movie 2). The fluorescent signal from the nuclei enabled the precise detection of individual cells within the tubular space (Fig. 4b; Supplementary Fig. 8a; Supplementary Movie 3). Moreover, we developed a Fiji code to virtually open the tube along its circumference into a planar equivalent (Supplementary Fig. 8b), such that we can quantify the collective movement on out-of-plane curvature.

First, we investigated the displacement of the LC in the TCSs due to the critical role of leader cells in regulating collective cell migration[48] and the relevance of different epithelial structures at the ductal tips (Fig. 2a and see above). As shown in Fig. 4c, the TCSs demonstrated linear progression with time. Interestingly, the displacement–time graphs for the larger TCSs ($\geq 100$ μm) exhibited a steeper slope than the smaller ones ($\leq 75$ μm) (Fig. 4c; Supplementary Fig. 8c), indicating that the TCSs under the highest constraint progressed with the slowest speed (see also Fig. 4d). Upon computing the average velocity of the cell front ($\overline{v_f}$) over the same period of time, we discovered that $\overline{v_f}$ for smaller TCSs ($\leq 75$ μm) increased as a function of tubular diameter up to about four-folds and plateaued in TCSs of 100 μm and above. This observation is in sharp contrast to that on 2D confined strips[12] and cylindrical wires[25], where $\overline{v_f}$ decreases as the constraint reduces. The fact that tissue expansion speeds were lower for microtube diameters <100 μm could be related to cell jamming in these tubes due to higher cell densities and the additional plugging of the smallest microtubes by multilayered tissue structures[49–51]. We also noticed that $\overline{v_f}$ of TCSs $\geq 100$ μm was comparable to that on flat tracks with a width larger than 100 μm[12–18]. This could be due to large TCSs having similar architectures (Figs. 2, 3 and see above) with epithelia on the broad 2D strips, and thus, their migratory pattern may resemble the planar counterparts. The $\overline{v_f}$ vs. diameter trend with increasing microtube diameter was robust as experiments using wild type (WT) MDCK (Supplementary Fig. 9) and another epithelial cell line, MCF-10A (Supplementary Fig. 10) yielded similar results as that for H1-GFP MDCK, suggesting a shared behavior between epithelial tissues under such conditions.

**Velocity fields and migration phenotypes of epithelial TCSs.** The previous observation motivated us to ask whether the variation of cell edge migratory velocity in different-sized TCSs arose from the differences in local and global velocity fields within the TCSs. The virtually opened tube images (Supplementary Fig. 8b) allowed the direct use of particle image velocimetry (PIV)[52] to map the velocity field (Fig. 5a), and in such images, x-axis represents the circumference of the tube, while y-axis denotes the tube length. Velocity vectors (arrows) in TCS of 25 μm in diameter revealed the presence of separate groups of cells migrating in opposite directions along the tubular long axis at a given instant (Fig. 5a, left), suggesting contraction–relaxation modes of migration. Similar behaviors had been reported previously[12] as a mode of MDCK monolayer migration caused by the spatial constraint on flat single cell-wide strips. However, the "tug-of-war" within the cell sheet on the highest curvature resulted in

much more prominent contracted–relaxed periods in the migratory patterns (Supplementary Movie 4). The periodic halt in the TCS advancement may explain the slow progression of the TCSs (Fig. 4c, d). In contrast, epithelial TCSs of 100 μm exhibited high longitudinal migration speed throughout the microtube (Fig. 5a, middle; Supplementary Movie 5). These high instantaneous velocities (~0.8 μm/min, deep red arrows) were observed at the leading front of the TCS as well as in the following cells that are 5–6 cell lengths behind. Furthermore, in the largest microtubes (250 μm), several cohorts of cells migrated in different directions (Fig. 5a, right, white arrows). Groups of velocity vectors representing high or low instantaneous velocity as well as ephemeral velocity vortices were noted in these TCSs (Supplementary Movie 6), resembling flat epithelial sheet behaviors.

Variation in the velocity fields indicates the alternation in coordination within the cell sheets. Indeed, we quantified the spatial velocity correlation lengths in both longitudinal and circumferential directions increased up to $\Phi > 100$ μm and plateaued at ~120–140 μm (Fig. 5b), which agrees with the values of 120–200 μm for planar unconfined substrates[12]. As expected, reducing confinement and curvature leads to the epithelial dynamics resembling the one observed on 2D unconfined surfaces. Moreover, the change of correlation lengths in both directions reminisces the $\overline{v_f}$ vs. diameter pattern (Figs. 4d, 5b), a phenomenon not seen in the 2D equivalents[12]. In addition, this is consistent with our previous observation of transition in the TCS architectures when the diameter increased beyond 75–100 μm (Figs. 2, 3). Consequently, it appears that tubular confinement/curvature is sufficient for both modulating the dynamics of epithelial cell–cell interaction and inducing alignment of cells in long distance. To confirm this hypothesis, we calculated the order parameter of the velocity, $S_v = <\cos(\phi)>$, where $\phi$ is the angle between the local velocity vector and the tube long axis, (Methods) and the results demonstrated a similar incremental trend which plateaued at ~0.80 when $\Phi > 100$ μm (Fig. 5c). This is again in contrast with planar confinement where $S_v$ decreases from a high value of 0.92 to ~0.80 as the width of the stripes increases[12]. We attribute the much lower order parameter in the smallest microtubes to the coexistence of the forward and backward velocity vectors, whose $S_v$ cancels one another (Fig. 5a, left). In the case of intermediate-sized microtubes ($\Phi = 75$ and 100 μm), PIV analysis demonstrated a velocity field with many vectors parallel to x-axis (i.e., large $\phi$, Fig. 5a, middle). This x-component of the velocity vector indicates that cells can migrate in the circumferential direction (circumferential swirling) and consequently, contributing to lower $S_v$. For the large microtubes (>100 μm), the TCSs exhibited similar order parameter and velocity field patterns as epithelia on wide flat stripes[12]. Although the transient vortices and circumferential migration frequently appeared, these were limited to small regions (mostly in the leading edge) of the sheets and the majority of cells migrated along the tubular long axis, bringing about a high $S_v$. Altogether, we conclude that the tubular confinement with curvature has a profound influence on the collective epithelial migration, which is different from that of conventional 2D flat constraint.

To further elaborate the dynamics of the TCS coordination, we investigated the variation in the velocity attained by the cell sheets in time and space. To quantify this, we plotted kymographs of the y- and x-components of the velocity averaged over the circumferential direction (Fig. 5d, e). Thus, the longitudinal vector (y-component) kymograph shows the spatiotemporal distribution of the local velocity in the direction of the planar surface (zero curvature), while the circumferential (x-component) kymograph presents the motion for an out-of-surface curved substrate. In highly confined tube, the alternate hot color (red/yellow) and cold color (blue/green) streaks with a period of

~60–80 min (Fig. 5d; Φ = 25 μm) were seen in the longitudinal vector kymograph, whereas the average transverse velocity was negligible over a long period (Fig. 5e; Φ = 25 μm). While the red/yellow streaks represented the overall forward movement of the cells that corresponded to the TCS progression, blue/green ones accounted for the halts or low speed backward motion, which could be attributed to the relaxation of the tissue. This confirms the contraction–relaxation modes under high confinement. In

contrast, in loose constrictions >75 μm, color coded for high x-axis velocity (0.6–0.8 μm/min) mainly localized in the frontier region and spanned more than 100 μm deep into the TCSs (Fig. 5d, middle and right). Such velocity field kymograph in these microtubes agrees well with large-scale coordination among the cells. The difference between the intermediate (Φ = 100 μm) and large (Φ = 250 μm) microtubes was reflected by the high circumferential velocity in the former, but much less

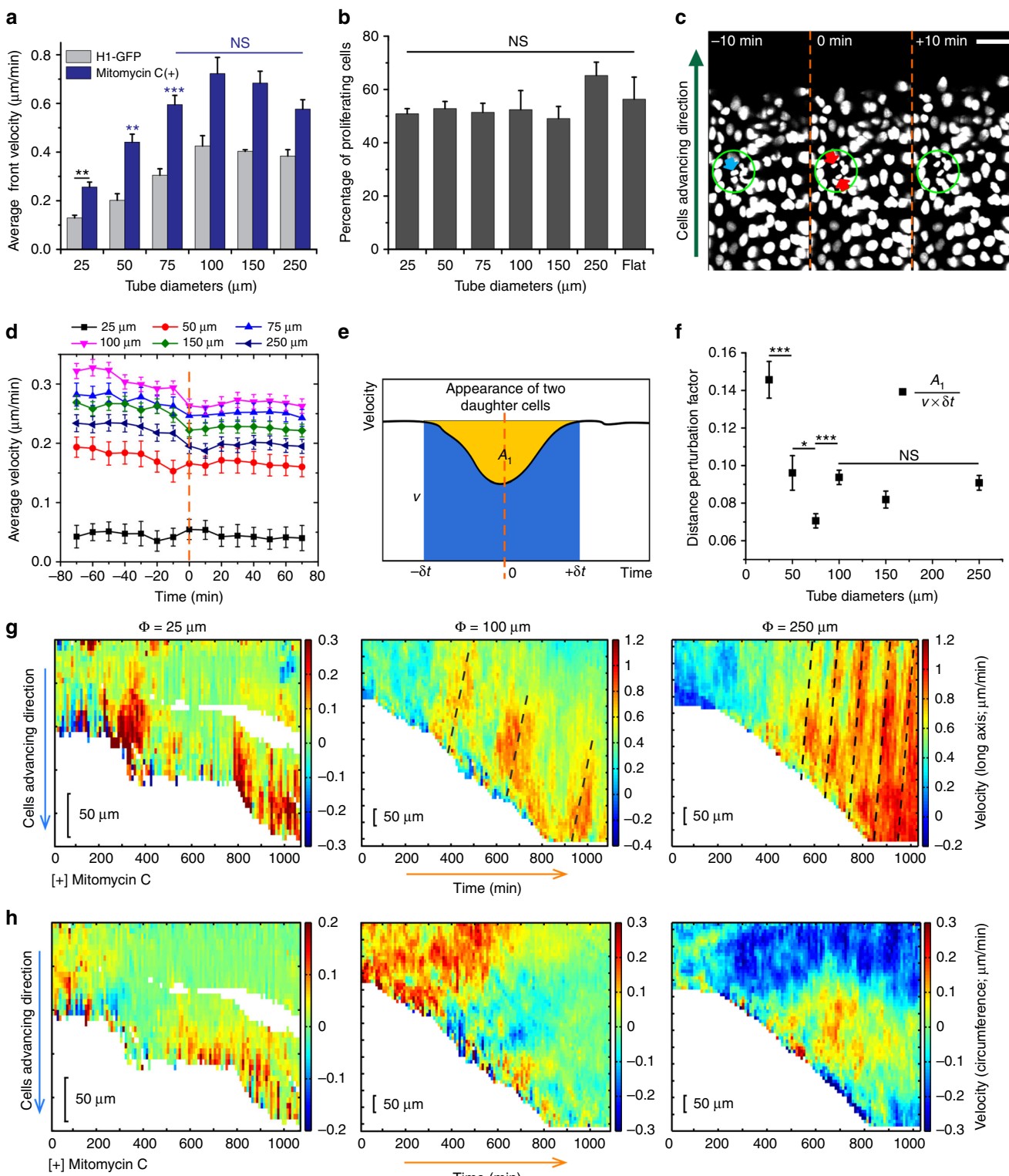

intense in the latter (Fig. 5e, middle and right). Furthermore, the plateau in the large microtube corresponded to the duration it took for the tissue to form a complete TCS, before fast advancement with constant forward progression was achieved (Fig. 5d, e, right).

**Influence of cell proliferation on migration velocities**. During the migration, cell division adds mass to the expanding TCSs but can also create disturbance to their mobility[53] in packed conditions. To investigate how cell division impacted tissue expansion, we first inhibited cell proliferation in MDCK tissues by mitomycin C treatment and found that this increases $\overline{v_f}$ in all conditions, but maintained the $\overline{v_f}$ vs. diameter trend (Fig. 6a; Supplementary Fig. 9), showing that cell divisions lowered tissue speed. To understand why this is the case, we first checked that the percentage of dividing cells in the tubes (over total number of cells in each tube) were similar across all tube diameters through EdU staining of proliferative cells[54] (Fig. 6b). Next, we measured the average velocity in the longitudinal tube direction within a small ROI (radius of ~25 μm) centered on cell division events (Fig. 6c), before (defined as negative time), during (defined as $t = 0$ min), and after division over a period of $\pm 70$ min ($t = 0$ min is defined as the first instance when the two daughter cells were observed). The individual ROI curves for each microtube size were further averaged, and these final curves showed that the local average longitudinal velocity always transiently reduced in magnitude around the division event (Fig. 6d, within $\pm 30$ min), before returning to the initial velocity values unperturbed by the division events (Fig. 6d, −70 and +70 min). As previously suggested, cell divisions can lead to tissue fluidization and increased viscosity[55,56] that slow down local velocity. To further understand why tissue speed was lower for smaller microtubes from the aspect of cell division, we quantified the magnitude of the velocity disturbance due to division (Fig. 6d) by calculating the area of the dip in the velocity curve over a certain period around division (Fig. 6e, yellow region, $A_1$) divided by the distance that should be traveled over the same period if there was no division (Fig. 6e, sum of the area of yellow and blue region, from $-\delta t$ to $+\delta t$) (Methods). This quantity, which can be understood as the percentage of tissue local distance perturbed by division, showed that the perturbation magnitude increased for smaller microtube diameters ≤75 μm up to nearly 15% per single division (Fig. 6f), which can be expected as smaller microtubes comprised of fewer cells along the circumferential direction and should be more easily perturbed by each division event. The fact that smaller microtubes are perturbed more by division events is also consistent with our finding that the speed increase in division inhibited tissue (mitomycin C treatment) compared to normal tissue is higher for smaller microtubes, quantified by the speed ratio $= \overline{v_f}$ without proliferation$/\overline{v_f}$ with proliferation

(Supplementary Fig. 11). Altogether, the results showed that the lower tissue speed trend in smaller microtubes was mainly influenced by the higher velocity perturbation in these tubes and not due to the difference in percentage of dividing cells in different microtube sizes.

We further generated velocity kymographs for the TCSs treated with mitomycin C. Consistent with the tissue front speed measurements, mitomycin C treatment enhanced the velocity along both the longitudinal and circumferential axes (Fig. 6g, h). Notably, the kymograph of longitudinal vector for small microtubes ($\Phi = 25$ μm) demonstrated analogous contraction–relaxation patterns with frequent detachment of the leader cells from the TCS (the blank regions in the kymograph, Fig. 6g, h, left, $n = 15/17$ from six independent experiments for 25 μm tube vs. $n = 0/16$ from six independent experiments for 250 μm tube). This may indicate the build-up of tension within the epithelial sheets in the TCS under high confinement with stress relaxation inhibited under these conditions. In TCSs ≥ 100 μm, we observed propagative waves opposite to the direction of migration (black dash lines in Fig. 6g, middle and right). These velocity waves traveled at ~3 and 10 μm/min inside 100 and 250 μm microtubes, respectively, corresponding to cell density alternations from 3 to 1 cells per 1000 μm². Such propagating velocity waves in coordinated migration have been identified in epithelial monolayers on 2D surfaces[57,58]. Our results are in line with the reported observation but also show that the existence of these propagative signals depends on cell confinement.

**Role of cell–cell junction in collective migration dynamics**. Since α-catenin is known to be a core mechanosensor of force transmission at cell–cell junctions[59–61], together with previous observations of the significant influence of α-catenin knock down (KD) on collective epithelial migration[12], we then addressed the role of cell–cell junctions by performing similar assays on α-catenin MDCKs. We observed that the loss of cell–cell junction stability often coincided with the loss of the multilayered epithelial organization at the leading front in the most constricted microenvironments (Fig. 7a), which could be due to the inefficiency of tension transmission between cells and thus the ineffective build-up of inward pointing forces (Supplementary Fig. 6d). Other epithelial architectures and organizations were also disturbed in α-catenin KD, e.g., the lack of longitudinally aligned actin stress fibers and more random cell orientation under the corresponding conditions (Fig. 7b, c). Moreover, due to unstable intercellular contacts, α-catenin KD MDCK cell sheets can sometimes break apart from the TCSs during ductal elongation (Fig. 7d; Supplementary Movie 7), reinforcing a crucial role of cadherin-based junctions in epithelial morphogenesis[62]. In terms of tissue dynamics, the loss of α-catenin in MDCK cells not

**Fig. 6** Cell division perturbs local velocity field more in smaller microtubes. **a** Average velocity of cell front of mitomycin C-treated H1-GFP MDCKs in tubes of different diameters (25, 50, 75, 100, 150, and 250 μm, from left to right), ($n = 6$ from three independent experiments in each condition) in comparison with that of H1-GFP MDCKs (taken from Fig. 4d). **b** Tubular confinement does not affect MDCK proliferation. **c** Representative images of cell nucleus (H1-GFP MDCKs) in a 50 μm diameter tubular tissue moving upward (green arrow). Green circle shows fixed region (radius ~25 μm), where velocity field vectors are averaged to determine the perturbation by a cell division event at $t = 0$ min, defined as the first instance where two daughter cells (red arrows) emerged from a dividing cell (blue arrow). Scale bar: 50 μm. **d** Average velocity in fixed region related to division as a function of time, further averaged over many division events. (25 μm: $n = 79$, 50 μm: $n = 130$, 75 μm: $n = 166$, 100 μm: $n = 286$, 150 μm: $n = 311$, 250 μm: $n = 348$, from two independent experiments per condition). **e** Schematic showing how the distance perturbation factor is calculated from the average velocity curves in **d** for a defined radius of fixed region, $r$. **f** Average distance perturbation factor as a function of tube diameter, where the factor is calculated for different pairs of parameters ($r$, $\delta t$), with $r = 18.6$, 21.1, 23.6, 26.0 μm and $\delta t = 10$, 20, 30 min. **g, h** Kymograph of the average longitudinal (**g**) and circumferential velocity (**h**) of mitomycin-treated H1-GFP MDCK TCSs migrating in different microtubes (from left to right: 25, 100, and 250 μm). Black dash lines indicate velocity waves that propagate opposite to the collective migration direction. $t$-test, *$P < 0.05$, **$P < 0.01$, ***$P < 0.001$, NS non-significant, for each condition, $t$-test has been performed between each microtube diameter and 25 μm, unless otherwise indicated by line. All data are presented as mean ± s.e.m.

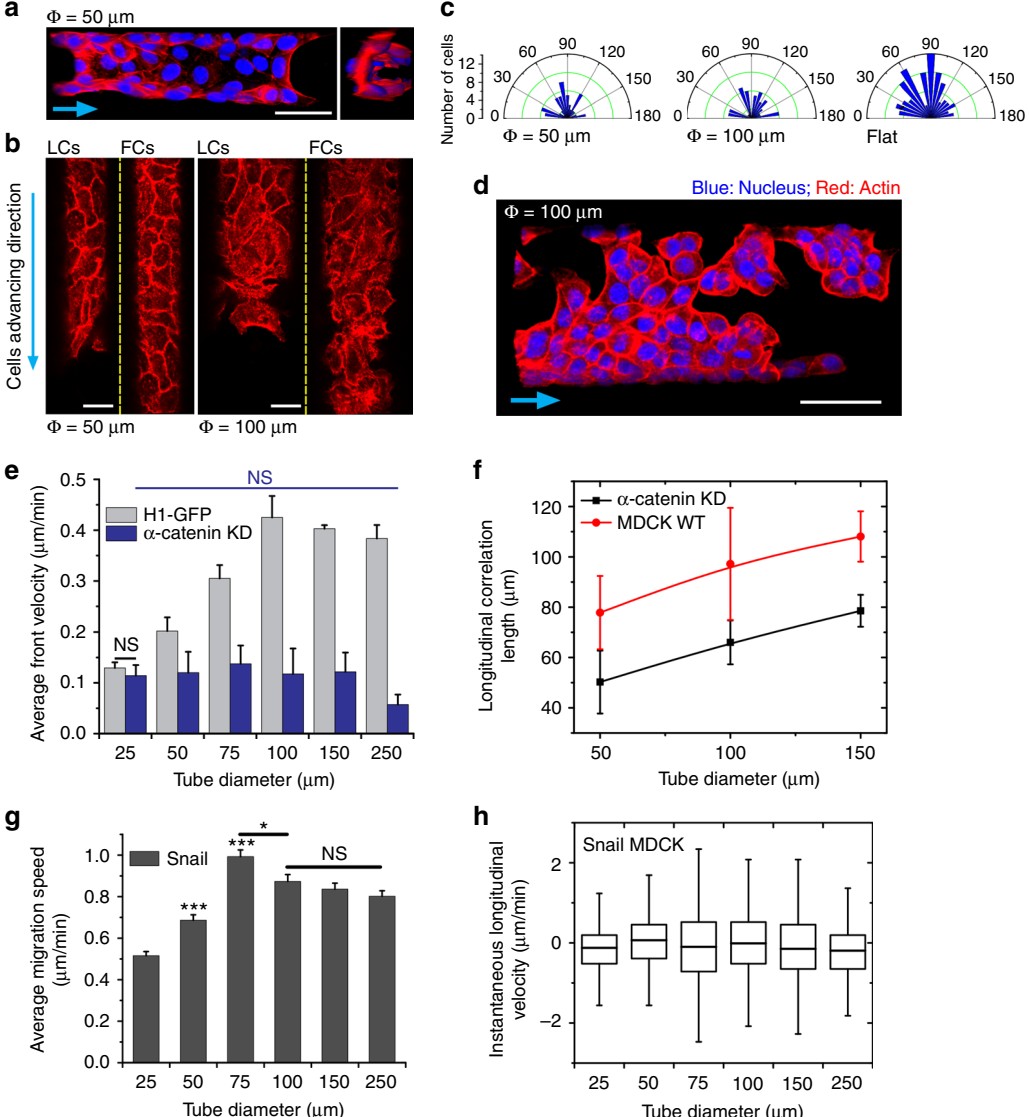

**Fig. 7** Cell–cell junction has significant influence on TCS migration. **a** 3D reconstructed fluorescent images of phalloidin (red) and DAPI (blue)-stained α-catenin KD MDCK TCSs in a microtube of 50 μm. Left, side view, and right, cross-sectional view showing the hollow lumen structure at the ductal tip. **b** Fluorescent images of basal actin stress fibers in different α-catenin KD MDCK TCSs. **c** Polar graph plotting the histogram of α-catenin KD MDCK cell orientation distribution under different confined conditions (50 μm microtube, 100 μm microtube, and flat substrate (from left to right)). **d** 3D reconstructed fluorescent image of α-catenin KD MDCK TCS (phalloidin: red and DAPI: blue) in a microtube of 100 μm in diameter. **e** Average velocity of cell front of α-catenin KD MDCKs in tubes of different diameters (25, 50, 75, 100, 150, and 250 μm, from left to right), (n = 6 from three independent experiments in each condition) in comparison with that of H1-GFP MDCKs (taken from Fig. 4d). **f** Longtudinal velocity correlation length for α-catenin knock down and WT tissue in phase contrast movies of microtubes of diameters 50, 100, and 150 μm (n = 6 from three independent experiment per condition). **g** Bar graphs showing average of instantaneous speeds and **h** the instantaneous longitudinal velocities along Snail-overexpressed cell tracks, for different microtube diameters, (25 μm: n = 11, 50 μm: n = 12, 75 μm: n = 16, 100 μm: n = 16, 150 μm: n = 16, 250 μm: n = 16 cell tracks of varying duration between 100 and 800 min, from two independent experiments per condition). Data presented as mean ± s.e.m. For each condition, t-test has been performed between each microtube diameter and 25 μm, unless otherwise indicated by lines, *P < 0.05, ***P < 0.001, NS non-significant. Scale bars: 50 μm in **a** and **d**, and 25 μm in **b**. Cyan arrows indicate the direction of collective migration

only significantly reduced $\overline{v_f}$ in all diameters, but also changed the $\overline{v_f}$ vs. diameter trend as the $\overline{v_f}$ became consistent for all diameters (Fig. 7e). This indicates that such collective behaviors cannot be explained by purely physical contacts between cells but require proper cell–cell contacts, particularly cadherin-based adhesions, to be efficient. Consistent with the low speeds in α-catenin KD tissue, the average velocity spatial correlation length for these tissues were smaller than those for WT tissue in microtubes of different sizes showing that they were also less coordinated in movement (Fig. 7f; Methods), indicating that collective dynamics remained crucial to explain cell expansion into the tubes.

Moreover, MDCK cells overexpressing Snail transcription factor that downregulates E-cadherin perturbed the collective cell behavior even more, and these cells failed to form TCSs even in the most constrictive microtubes as they exhibited single-cell migratory behaviors (Supplementary Movie 8). The instantaneous Snail-overexpressed cell speeds along their single-cell tracks increased with increasing microtube diameters, and reached a saturation point at microtube diameters ~75–100 μm (Fig. 7g), mirroring the trend found for tissue expansion speeds as a function of microtube diameter for collective tissues (Fig. 4d; Supplementary Figs. 9, 10a). However, the instantaneous velocity for Snail-overexpressed cells

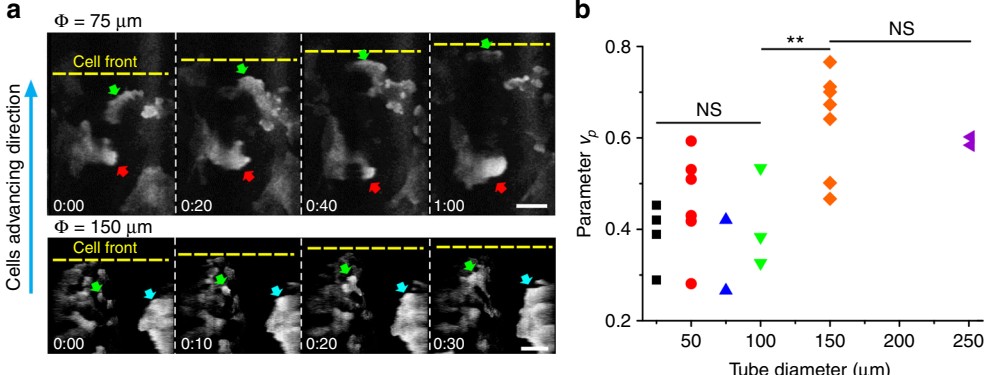

**Fig. 8** Forward polarization is better established in bigger microtubes. **a** Representative images of PBD-YFP cells in a 75 μm diameter tubular tissue moving upward (cyan arrow). Green (red) arrows show the active PBD zones as marker of lamellipodia protrusion in (opposite to) the direction of tissue front (yellow dashed line) expansion. Time is given in h:min, scale bars: 20 μm. **b** Calculation of the average lamellipodia persistence parameter, $v_p = \sum_i v_i / \sum_i |v_i|$, as a function of tube diameter, where $v_i$ are all the velocity vectors falling into the zones marked by the active PBD signal (see Methods for how the zones are determined). Each point denotes an independent experiment. $t$-test: $**P < 0.01$, NS non-significant

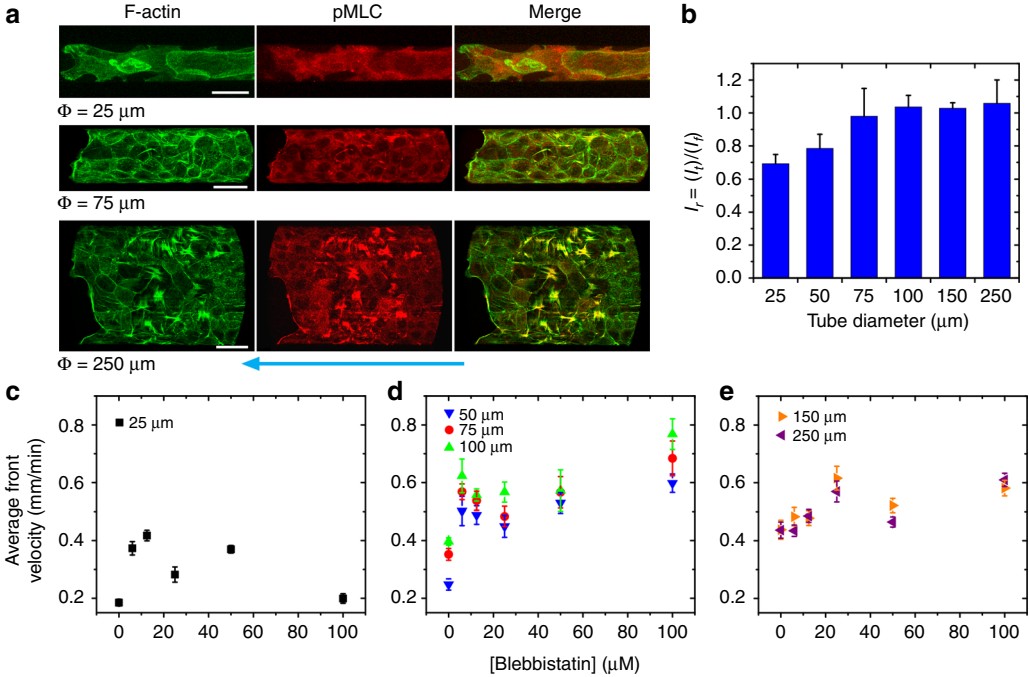

**Fig. 9** Acto myosin contractility is different in various sized TCSs. **a** MDCK-WT TCSs fixed and stained for actin (green) and pMLC (red). Scale bars, 25 μm for top panel and 40 μm for middle and bottom panels. Cyan arrow indicates the direction of collective migration. **b** Relative fluorescent intensity, $I_r$, (pMLC, red fluorescent intensity in TCSs as in **a** relative to flat condition, see Supplementary Fig. 12) as a function of TCS diameters ($n = 6$ from three independent experiments in each condition). **c-e** Tissue expansion speed as a function of blebbistatin concentration can be separated into three groups. **c** About 25 μm with initial speed increase then decrease at higher blebbistatin concentration ($n = 7$ from three independent experiments in each condition). **d** About 50-100 μm with continuously fast increase in speed with increasing blebbistatin concentration ($n = 6$ from three independent experiments in each condition). **e** About 150 and 250 μm with slow increase in speed with increasing blebbistatin concentration ($n = 6$ from three independent experiments in each condition). Data are presented as mean ± s.e.m.

along the microtube distributed mostly symmetrically about the zero value for all microtube sizes (Fig. 7h), which would lead to no average net movement anywhere, unlike other tissues which persistently migrated forward in the microtubes. Taken together, our results indicate the importance of cell–cell adhesion for the regulation of coordinated tissue migration and the maintenance of epithelial integrity and net tissue speed during tubulogenesis.

**Lamellipodia protrusion under various tubular confinement.**
As forward polarization induced by lamellipodia protrusion at the tissue front[63] or cryptic lamellipodia extension within the epithelium monolayer[64] play an important role in collective migration, we first checked how well cells polarized within the microtubes of different sizes based on the velocity and direction of lamellipodia-based protrusions. To this end, we monitored the basal sections of cells in tubular tissues (virtually opened as described above) stably expressing YFP-tagged p21-binding domain (PBD-YFP) probe of activated Rac1 and Cdc42[65], using the bright, activation zones as markers of lamellipodia structures (Fig. 8a, arrows). In general, the direction of lamellipodia protrusion correlated with the forward movement of the TCSs, but

importantly, cells tended to portray numerous lamellipodia protrusion opposite to the movement of the tissue front (Fig. 8a, red arrows), which occurred more frequently in the smaller microtubes. To quantify this observation, we plotted a lamellipodia persistence parameter, $v_p$ defined as $\sum_i v_i / \sum_i |v_i|$, where $v_i$ is the velocity associated with the direction of lamellipodia protrusion and $|\ldots|$ denotes the absolute value (Method). This parameter would be equal to 1 (−1) if all the lamellipodia protruded in the same (opposite) direction as the tissue front movement, while its absolute value would decrease if lamellipodia protrusion in both directions occurred. Consistent with our observations (Fig. 8a), the persistence parameter was smaller for microtubes with diameter ≤100 μm (Fig. 8b), suggesting that tissue forward polarization was poorer in such tubes, in line with the measurement of taller cell height (Fig. 2c), lower tissue speeds (Fig. 4d; Supplementary Fig. 9), and contraction–elongation modes of migration (Fig. 5a, d). The fused multilayered structure observed at the migration front in smaller microtubes (Fig. 2a; Supplementary Fig. 4) could also explain the less pronounced forward polarization there.

**Contractility difference leads to migration speed variations.**
Then we investigated the role of acto myosin contractility, by studying phosphorylated myosin light chain (pMLC) distribution in the tubular tissues. pMLC signal was found to be diffuse, non-colocalizing with actin stress fibers and weak in intensity ($I_t$) in the smallest tubes (diameter <50 μm, Fig. 9a, upper panel) compared with that on flat substrate ($I_f$, and Supplementary Fig. 12). In contrast, we observed stronger signals of pMLC intensity that colocalized with actin stress fibers in larger tubes (Fig. 9a, middle and lower panels), similar to those on flat surfaces. The changes in pMLC fluorescent signal as a function of TCS diameter was quantified by the relative fluorescent intensity, $I_r = I_t/I_f$ (Fig. 9b) and the plot demonstrates an increasing trend from low $I_r$ in small tubes to high values in large ones, where the $I_r$ plateaus. This indicated that contractility level was lower in smaller tubes with higher confinement/curvatures.

In order to gain a further understanding about contractility in TCS collective migration, we studied tissue front speed ($\overline{v_f}$) as a function of varying blebbistatin concentrations (from 6 to 100 μM), which inhibited acto-myosin contractility to different degrees. Results showed that $\overline{v_f}$ varied non-monotonically with blebbistatin concentration[12,55]. The contractility behavior can be separated into three distinct groups based on microtube size, for $\Phi = 25$ μm, diameter ≤50–100 μm, and diameter > 100 μm (Fig. 9c–e, respectively), confirming the differences in contractility between smaller and bigger microtubes. Epithelial cell velocities in all microtubes increased in general with increasing blebbistatin concentration up to 50 μM (Fig. 9), possibly due to a decrease in tissue elasticity that allows the tissue to extend more easily[55,66]. Importantly, only tissue speed in the smallest tube (25 μm) presented a drastic drop when blebbistatin concentrations increased more than 50 μM, indicating the myosin-II contractile activity levels for such tubes were lowest and, such that they are the first to be depleted of activity. This is consistent with the pMLC results showing that acto myosin activity in smallest TCSs was initially low. Overall, the analysis of pMLC staining and blebbistatin drug treatment results suggested that acto myosin contraction decreased with higher tube curvature.

## Discussion

The formation of epithelial tubules and cysts require a population of cells to coordinate their behaviors across space and time to interact with the microenvironment. While epithelia demonstrate extensive, collective migratory responses to external geometric signals, in-depth understanding of how constraint and out-of-plane curvature regulate tissue migration, including cell rearrangement and organization, tissue velocity field, and its order, is mostly lacking to date. 2D flat assays have been previously evidenced for the guidance of in-plane curvature and confinement in wound healing[24] and epithelial dynamics[12]. In addition, when plated on a micro-cylinder of positive curvature, epithelia organize into tubular architecture of an inverted polarity and show a curvature-induced EMT[25]. However, both systems failed to recapitulate the processes of the formation of a hollow epithelial lumen. Though conventional soft lithography method has enabled the production of lumen networks, it is commonly limited by a rectangular cross-section[32]. While most of our current understanding about epithelial tubulogenesis comes from 3D gel approaches[8,10], very little information about the mechanisms of migration and tissue–environment interaction is obtained by such systems. In contrast, even though the TCSs inside the microtubes may be at best considered as pseudo-3D due to the constrained movement on a curved surface, our assay mimics aspects of epithelial architecture and collective migratory dynamics leading to tubule formation that are highly pertinent to in vivo situations.

We have demonstrated that there is a transition in epithelial organization and migratory pattern due to the increasing tubular spatial constraint and/or curvature. In smaller microtubes with high surface curvature, cells are taller and thus TCSs can exhibit multilayered, fused structures that plug the smallest microtubes. This is in line with the observation that cells spread less on negative curvature surfaces, which can be due to a plausible interfacial stress that increases with the curvature. Due to this unique structure, cells could be more jammed in the smaller tubular tissues, which also consistently portrayed poor forward polarization, marked by haphazard lamellipodia protrusion directions. On flat surfaces, MDCK cells within confluent tissues started to transition from a more directed migration to a more jammed, diffusive motion with lower speeds around the cell density threshold of 3–4 cells/(1000 μm²)[49–51]. In contrast, the density threshold that separated smaller microtubes (≤100 μm diameter, with lower tissue speeds) from bigger microtubes was measured to be ~2 cells/1000 μm² (Fig. 2b; Supplementary Fig. 6b), which suggested that surface curvature can shift the jamming transition to lower cell densities. The possibility of jamming in smaller tubes also coincided with the observation of backward–forward motion and swirling[67,68] that predominantly happened in microtube diameters ≤100 μm (Fig. 5a, d, e).

Further, based on contractility results from pMLC staining and blebbistatin treatment, tubular tissues in the smallest microtubes can be deduced to have smaller active tissue propulsive stress consistent with less pronounced forward polarity, and could also have smaller traction forces. Tubular tissues in smaller microtubes were also found to experience more cell division-induced perturbations to their forward movement probably due to their smaller size. All these contributed to a mechanistic understanding of the decrease in tissue expansion speed in smaller microtubes. Importantly, the unique TCS migration strongly depended on the collectivity of the cells provided by a strong cell–cell junction, which could increase coordination between cells to induce better forward polarization. Consistent with this view, α-catenin KD in the MDCK cells portrayed lower tissue expansion speed in all microtube sizes and frequently disintegrated, indicating that cell–cell adhesion is an important driving force for tubule migration in highly constricted environments. On the other hand, the more coordinated velocity patterns in the larger microtubes suggested that once the confinement is reduced, cells at the leading edge of the tube can transmit tissue stress much deeper into the tissue bulk. All these results showed that the emergence of different modes of collective cell migration in tubular

confinement is in sharp contrast with the effect of 2D geometric constraints seen earlier[12] and the positively curved cylindrical surface[25].

The effects of tubular constraints that we have observed on key migration features such as migration velocity, order parameters, and velocity field distribution reflect the impact of spatial confinement and curvature on epithelial lumen development. Along this line, the evidence of frequent detachment of front cells from the groups without proliferation in very narrow microtubes may be reminiscent of EMT[19,56]. Cell detachment events may result from a tug-of-war between cell–substrate and cell–cell interactions and as such, due to an increased tensile stress within the tissue under mitomycin treatment since cell proliferation can be seen as a source of stress dissipation[56]. It will be appealing to further investigate whether extreme constrictions, occurring during certain pathological conditions, could lead to aberrant morphogenesis.

The easy combination of our microtube platform with high-resolution microscopy and simple construction have helped us to interrogate the pseudo-3D cellular behaviors and data with 2D technique and analysis. Such approach could be further generalized and extended to more complicated 3D systems. With all these findings, we demonstrate the profound influence of spatial confinement on epithelia and provide an in-depth characterization of collective cell migration on tubular negative curvature reminiscent of in vivo tubulogenesis.

## Methods

**Fabrication of circular microchannels.** Smooth metal wires (typically made from copper or platinum) were aligned in parallel just one to two microns above a silicon wafer with a precise stage, which can be used to control the distance between each wire and the silicon wafer surface. A freshly mixed PDMS (a mixture of Sylgard 184 silicone elastomer base and Sylgard 184 silicone elastomer curing agent, 10:1 by weight) was poured on top of the silicon wafer covering the metal wires. The whole setup was then transferred into an 80 °C oven for 2 h or left at room temperature for 24 h to cure the PDMS. After the polymerization of the PDMS, the metal wire was pulled out from the PDMS microtubes during a sonication process in an acetone solution, which would extract unreacted elastomer curing agent and cause slight swelling in the polymer—loosening the PDMS–metal contact. The detached PDMS microtubes were then baked in an 80 °C oven for 30 min to remove any acetone remnant. The PDMS blocks containing straight microtubes were cut into small pieces with tubular microchannels of ~1 mm in length. The microchannels then had two openings, and the short length allowed the efficient diffusion of nutrients into the microtubes.

To allow collective epithelial cell migration into the microtubes, one piece of the PDMS block containing microtubes coated with fibronectin was placed in front of a migrating MDCK cohort with one opening facing the moving cell sheet. Due to the very narrow step (~1–2 μm between the microtubes and the substrate, see above) MDCK cells were able to crawl into the microtubes with intercellular cohesion. The whole setup was immersed into complete medium and mounted on microscope for imaging.

The fabrication of bowl-shaped structures with negatively curved surfaces was done with SU8 coating on silicon wafer and common lithography methods, which eventually formed dome structures. PDMS was cast over the domes to create the bowl structures.

**Cell culture and imaging.** MDCK-WT (Madin-Darby canine kidney wild type) cells, MDCK-H1-GFP (stable cell line transfected with H1-GFP), MDCK-PBD-YFP (stable cell line expressing YFP-tagged p21-binding domain probe of activated Rac1 and Cdc42, kindly provided by Fernando Martin-Belmonte, Universidad Autónoma de Madrid), MDCK-E-cadherin-GFP (stable cell line expressing E-cadherin-GFP), MDCK-Snail (cells overexpressing Snail transcription factor), and MDCK α-cat KD (stable cell line with α-catenin KD) were cultured in DMEM (Life Technologies) supplemented with 10% fetal bovine serum and 1% penicillin/streptomycin. MCF-10A cells (American Type Culture Collection, ATCC) were maintained in MEGM (Lonza) medium supplemented with Cholera toxin (100 ng/ml). Microtubes were coated with 50 μg/ml Fibronectin (Sigma-Aldrich) for 1 h, washed with 1× PBS and then cells were seeded close to the edge of the microtubes. Once the migrating cell sheets entered the microtubes, the sample was placed on the confocal microscope (Nikon A1R or Zeiss LSM 710) and z-stacks (1–2 μm per step) covering the entire volume of the tubes were imaged every 10 min. Time-lapse images were performed over a period ranging from 12 to 24 h. For inhibiting cell division, mitomycin C (Sigma-Aldrich, M4287) was added at a concentration of 10 μM, incubated for 1 h, washed away and replaced with fresh medium before imaging. For myosin-II inhibition, cells were treated with 6, 12.5, 25, 50, and 100 μM blebbistatin (Cayman Chemical, 13013) 30 min before imaging.

**Image analysis.** Virtual opening of the TCSs was performed after background subtractions on ImageJ. Time-lapse z-stacks of the tubes were first resliced to obtain the xz plane. A circle was fit to the circumference of the TCS, converted to line, straightened and resliced again to obtain a time-lapse image of virtually opened tube with the tube circumference (x-axis) and length of the tube (y-axis) as the new axes. This process was coded in ImageJ to help in converting a 3D time-lapse image into 2D and eased the process of analysis while conserving the relevant x, y, and z information. PIV was performed on the virtually opened tube images using PIVLab code[48] on MATLAB. Angle between the velocity vectors obtained from PIVLab and the correlation length of the migrating TCS were calculated using a MATLAB code. Average velocity of cell migration was calculated on ImageJ. Orientation index of the cells was also calculated on ImageJ by considering individual cells as ellipses ($n > 100$).

To analyze the velocity field vectors related to activated (bright) PBD zones as a marker of lamellipodia protrusion, PIV was first performed on the virtually opened, basal layer of the original confocal images of the PBD tissue. From the open images of PBD signals, the bright zones were specified automatically using the Auto Local Threshold plugin in ImageJ, with the midgrey and radius = 30 pixels options (resolution of images were 0.62 μm/pixel). These selected zones were visually compared with the original open images to verify that they mirrored reasonably the bright PBD zones, and only the velocity vectors falling within these regions were selected for further analysis. From these velocities, a lamellipodia persistence parameter, $v_p = \sum_i v_i / \sum_i |v_i|$ was calculated, where $v_i$ is the velocity associated with the direction of lamellipodia protrusion and $|\ldots|$ denotes the absolute value. This parameter would be equal to 1 (−1) if all the lamellipodia protruded in the same (opposite) direction as the tissue front movement, while its absolute value would decrease if lamellipodia protrusion in both directions occurred. A parameter closer to 1 would indicate that the cells within the tubular tissue are better polarized in the front–back direction.

To quantify the disturbance that cell division events cause to the tissue expansion dynamics, the average longitudinal velocity vectors within a fixed circular region (of radius, r) centered on each division event (between two daughter cells that first emerged from the mother cell) were determined before (starting from −80 min), during ($t = 0$ min), and after (up to +80 min) the division (Fig. 6c, d). Such average velocity curves were further averaged for all division events, centered on their respective $t = 0$ min. A sharp dip in this average velocity curve around $t = 0$ min, indicative of the division-generated disturbance to local forward velocity fields, is seen superposed on a slowly decreasing velocity trend with increasing time at long-time scales (Fig. 6d). To quantify the dip as a way of numerating the disturbance, we first leveled the slow tilt in the average velocity curve and approximated it as a horizontal, straight line. The area bounded by this straight line with the leveled average data, over a certain duration (−δt to δt around $t = 0$ min, Fig. 6e) quantifies the perturbation to the distance cells would travel if there were no perturbation (positive if velocity drops, negative if velocity increases). A distance perturbation factor was quantified by normalizing this area with the area under the straight line within the same duration, such that data from different microtube sizes can be compared. To reduce bias, this factor was calculated for different pairs of parameters ($r$, δt) with $r = 18.6$, 21.1, 23.6, 26.0 μm and δt = 10, 20, 30 min, and all taken into account for statistics.

Phase contrast movies were used with α-catenin KD and Snail-overexpressing cells in microtubes as these cells have no fluorescent markers. To better approximate the cell position and velocity components in the circumferential direction of tubes in phase contrast movies that only show a 2D projection of these values, the formula $x_{\text{circumference}} = R \times \sin^{-1}(x/R)$ was used to make the positional conversion, where $x$ is the position in the circumferential direction measured in phase contrast images, $R$ is the microtube radius, with the coordinates taken to center around the midpoint of the tube axis taken as $x=0$. The correction to the circumferential velocity was calculated by differentiating this formula, obtaining $v_{\text{circumference}} = v / \sqrt{\left(1 - \frac{x^2}{R^2}\right)}$, with $v$ being velocity in the circumferential direction measured by PIV. The longitudinal parameters are correct as measured.

To quantify the relative pMLC fluorescent intensity, we first obtained a maximal intensity z-stack projection of the basal layers of original confocal images to avoid the curvature effect. The fluorescent intensity in TCSs ($I_t$) was then measured over a rectangular shape of $5 \times 50$ μm$^2$ along the tubular longitudinal direction and the intensity on flat surfaces ($I_f$) was measured within the same ROI. The relative pMLC fluorescent intensity ($I_r$) was defined by the following formula: $I_r = I_t/I_f$.

**Sample fixation and staining and confocal microscopy.** Cells were grown and allowed to form tubular cell sheets inside microtubes. After the TCS progressed into the microtube for 24–48 h, the samples were fixed with 4% paraformaldehyde for 15 min or −20 °C methanol for 5 min and then permeabilized with 0.2% Triton X-100 or 0.025% saponin for 15 min. Actin cytoskeleton was visualized after staining with Alexa-488 (Invitrogen, A12379, 1:250) or Alexa-568 (Invitrogen, A12380, 1:250) labeled Phalloidin for 1 h. To assess cell polarization, samples were incubated overnight in 2 μg/ml anti-Gp135 (DSHB, Q52S86) and then stained with 1:250–500 dilution of Alexa Flour 488 secondary antibody (Invitrogen, A11001).

To stain for pMLC, samples were incubated overnight at 4 °C with the anti-pMLC$^{T18+T19}$ antibody (Cell Signaling, #3674s, 1:50–100) and then stained with Alexa Fluor 568 secondary antibody (Invitrogen, A11011, 1:250) at room temperature for 1 h. Cells were then mounted with anti-bleaching medium (Vector Laboratories, H-1000) and z-stacks of the cell tubes were acquired on a confocal microscope (Nikon A1R or Zeiss LSM710).

**Calculation of the TCS parameters.** Cells in TCSs were considered as ellipses, and the orientation of the cell was calculated as the angle between the major axis of the cell and the long axis of the microtubes. The order parameter for actin filament was defined as cosine of twice the angle, $\theta$ of the filament with the tubular longitudinal axis. The overall actin order parameter for a certain TCS was calculated using the formula: $S_a = \langle\cos(2\theta)\rangle$. The average order parameter for velocity vectors in PIV mapping was defined by the following formula:

$$S_v = \langle\cos(\phi)\rangle$$

where $\phi$ refers to the angle which the vector makes with the y-axis in the virtually opened TCS 2D image. Therefore, $S_a = 0$ refers to an isotropic alignment, $S_a = 1$ refers to a perfect orientation along tubular longitudinal axis, and $S_a = -1$ refers to a perfect alignment vertical to the tubular long axis. On the other hand, $S_v = 0$ corresponds to a random distribution of velocity vectors in the TCS, $S_v = 1$ corresponds to an overall forward movement (vectors parallel to the microtube length and directed along the direction of the migrating cell sheet) and $S_v = -1$ corresponds to an overall backward movement (vectors parallel to the microtube length and directed opposite to the direction of the migrating cell sheet).

If two vectors are correlated, one of the vector values and directions can be predicted by knowing the other vector. Two vectors which are closer spatially to each other usually have higher correlation with one another, compared to two vectors further apart. The average correlation magnitude between two vectors that are separated by a distance $|\vec{r}|$ is given by the correlation function defined as[12]:

$$C_u\left(\vec{r}\right) = \left\langle \frac{\left\langle u^*\left(\vec{r}' + \vec{r}, t\right) \times u^*\left(\vec{r}', t\right)\right\rangle \vec{r}'}{\left[\left\langle u^*\left(\vec{r}' + \vec{r}, t\right)^2 u^*\left(\vec{r}', t\right)^2\right\rangle\right]^{1/2}} \right\rangle t$$

$$C_v\left(\vec{r}\right) = \left\langle \frac{\left\langle v^*\left(\vec{r}' + \vec{r}, t\right) \times v^*\left(\vec{r}', t\right)\right\rangle \vec{r}'}{\left[\left\langle v^*\left(\vec{r}' + \vec{r}, t\right)^2\right\rangle \left\langle v^*\left(\vec{r}', t\right)^2\right\rangle\right]^{1/2}} \right\rangle t$$

where $C_u$ and $C_v$ correspond to the correlation function in the directions that are vertical and parallel to the microtube longitudinal axis, respectively, $u^*$ and $v^*$ are the deviation of the velocity from the mean velocity in the directions vertical and parallel to the microtubular long axis, $\bar{r}$ is the vector of the coordinates, and $t$ is time. The spatial velocity correlation functions were computed by calculating the mean of the correlation coefficient over all the directions such that $C_u$ and $C_v$ are now functions of $\|\bar{r}\|$ (the norm of $\bar{r}$). The correlation length then refers to the distance ($\|\bar{r}\|$), where the spatial velocity correlation function first reaches zero.

**Code availability.** All codes will be available upon request from the corresponding authors.

**Data availability.** The data that support the findings of this study are available within the article and Supplementary Information or available from the authors upon request.

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

## Acknowledgements

We thank Dr. Fang Kong from SMART, Singapore for helping to design and manufacture the precise stage for the microtube fabrication as well as to make Supplementary Fig. 1, Wai Han Lau and Jun Liu from the microscopy core of Mechanobiology Institute Singapore (MBI) for imaging support, Hui Ting Ong and Dr. Hayri Emrah Balcioglu from MBI for help in programming and Dr. René-Marc Mège and Dr. Delphine Delacour from IJM, Paris, and Dr. Sham Tlili from MBI for helpful discussion. We thank Song Hui Tan and Bee Leng Tan from MBI laboratory facility for supporting in the lab work and Chun Xi Wong from MBI Science Communications Unit for help in making illustrations. We also thank MBI students—Yejun Wang, Zhe Wen, Yue Zhang, Anh Phuong Le, and Shreyansh Jain for support in the experiment. The MDCK cell lines were kindly provided by Sham Tlili, Shigenobu Yonemura, and W. James Nelson. The monoclonal antibody of anti-Gp135 developed by investigators was obtained from the Developmental Studies Hybridoma Bank, created by the NICHD of the NIH and maintained at The University of Iowa, Department of Biology, Iowa City, IA 52242. Financial supports from the Mechanobiology Institute seed grant, Ministry of Education's Academic Research Fund (AcRF) Tier 1 Grant (R-397-000-247-112), the European Research Council under the European Union's Seventh Framework Program (FP7/2007-2013)/ERC grant agreements no. 617233, the LABEX "Who am I?" and USPC-NUS funding are gratefully acknowledged.

## Author contributions

W.X., B.L. and C.T.L. conceived the project, W.X. designed the experiments. B.L. and C.T.L. supervised the study. W.X., S.S. and T.B.S. performed and analyzed all experiments. T.B.S. programmed codes for analysis. W.X. made the figures. W.X., S.S., T.B.S., B.L. and C.T.L. wrote the manuscript. All authors commented on and/or edited the manuscript and figures.

## Additional information

**Competing interests:** The authors declare no competing financial interests.

