## [Peer Review File · Nature Communications]

Reviewers' comments:

Reviewer #1 (Remarks to the Author):

In this manuscript, Xi et al. fabricate novel cylindrical microchannels to study collective cell migration. Overall, this is an interesting, albeit phenomenological, study. Unfortunately, the lack of any mechanistic insight(s) markedly diminishes the impact of this manuscript.

Major comments:

1. The authors should provide more insights on the role of contractility, cell proliferation and cell-cell junctions in the process of collective cell migration in cylindrical microtubes. Also, why does mitomycin C have such a strong effect on cell speed? Does cell proliferation increase in cylindrical microtubes? How does myosin activity change as the tube diameter increases? How does the loss of α -catenin induce loss of actin alignment? Why does loss of α -catenin decrease migration speed in larger diameter tubes and not in smaller tubes? Does loss of α -catenin "resemble" Snail overexpression? Although the authors have performed experiments with Snail overexpressing cells, they do not provide any quantitative analysis; at least, speed measurements are necessary.
2. Statistical analysis is missing throughout the manuscript.
3. The authors need to provide an explanation on why the height of the cells increase as the tube diameter decrease.
4. Are the authors' observations cell line specific? Experiments with additional cell lines are required to generalize findings.
5. Since the authors use PDMS (which is very stiff) to study collective migration, do they expect similar results under physiologically relevant stiffnesses? How does substrate stiffness affect collective migration in microtubes?
6. The authors use the same control cell line for the inhibitor, the knockdown and overexpression studies. The appropriate control cell line needs to be used for each experiment.

Minor comments:

1. Page 17 line 493, $\text{Cos}(2\varphi)$ needs to be replaced with $\cos(\varphi)$
2. Page 17 line 504, a description of C_u and C_v is missing.
3. Correlation length needs to be better explained.
4. Page 6 line 150 "is was then"- Please correct this.

Reviewer #2 (Remarks to the Author):

Up to now, few studies have looked at dynamical aspects of coordinated epithelial behaviors across space and time in non-planar geometries. To address these challenges, this manuscript uses a microtube platform for the study of collective epithelial dynamics that leads to the formation of

hollow tubules. The results are novel, interesting, and important. The manuscript suffers from a few typos and grammatical errors, but otherwise is very well-written. I have only a few comments that will need to be addressed.

Lines 71-74. These arguments fail to thread the needle. Yes, gel and matrix based 3D assays are more complex, and the geometry is self-assembled, but these situations are closer to the real biology. In your preparation you gain control of the curvature but move away from the biology. You should probably just say that.

Line 220: "As shown in Fig. 4d, this increased the average front edge velocity in all conditions." This is counter-intuitive, no? Is it due to an off-target effect. Some comment is warranted here is very much needed.

Line 228 "The inhibition of myosin-II through blebbistatin treatment increased the velocity in all tubes" This is counter-intuitive as well. Some comment is warranted.

Discussion: When you talk about swirling motions in space and time, I was surprised that you did not mention at all cell jamming effects and associated glassy dynamics that have been well described in the recent literature for flat culture. If you think that the dynamics here are unrelated to those phenomena, you should say so, and defend your assertion. And if you think that these effects are in play, you should say that.

Point-by-point response to the reviewers' comments:

First of all, we would like to thank the reviewers for their careful reviewing of the data presented in this manuscript and their constructive comments, which have allowed us to revise and improve the manuscript significantly.

Reviewer #1 (Remarks to the Author):

In this manuscript, Xi et al. fabricate novel cylindrical microchannels to study collective cell migration. Overall, this is an interesting, albeit phenomenological, study. Unfortunately, the lack of any mechanistic insight(s) markedly diminishes the impact of this manuscript.

Response:

We thank the reviewer for his/her thorough comments. As noticed by the reviewer, this manuscript was indeed intended to be mostly phenomenological initially, since there are only few studies which give a detailed account of cellular/tissue dynamics on curved surfaces, and we have overcome considerable technical difficulties in doing these experiments and measurements. However, we agree with the reviewer that adding mechanistic insights would significantly increase the impact of this work. In the revised version of the manuscript, we have incorporated new experiments and explanations which provide some mechanistic insights that better explain and quantify our experimental data.

Major comments:

1. The authors should provide more insights on the role of contractility, cell proliferation and cell-cell junctions in the process of collective cell migration in cylindrical microtubes.

Response:

As suggested by the reviewer, we have added mechanistic insights to explain the unique features in collective cell migration (CCM) in cylindrical microtubes, e.g. the trend where tissue expansion speed decreases with increasing confinement and curvature (reducing microtube size). We have not only addressed the roles of contractility, cell division and cell-cell junctions in this process, but have also added experiments on forward polarization of cells and tissue spreading on negatively curved surfaces, and linked the experimental results to several known simulation results in literature to gain a better understanding of the process. Besides improving current figures, we have included 4 new figures, Fig. 6-9, dedicated to the mechanistic understandings of the CCM process. Detailed response can be found below.

FORWARD POLARIZATION:

Firstly, we investigated the forward polarization in the tubular cellular sheets (TCSs) and found that bigger tubular tissues indeed portrayed more consistent forward lamellipodia protrusion (Fig. R1a, green and cyan arrows). However, in smaller TCSs, cells tended to portray numerous lamellipodia protrusion opposite to the movement of the tissue front (Fig. R1a, red arrows). In general, the direction of lamellipodia protrusion correlated with the forward movement of tissue in the microtubes, i.e., persistent lamellipodia extension in the same direction of migration indicates high tissue polarization and thus, faster tissue expansion speed, while haphazard lamellipodia protrusion direction indicated otherwise. We quantified these observations using a lamellipodia persistence parameter (v_p , Fig. R1b) as presented in the new manuscript in page 14 and have included a detail discussion in the section of "*Haphazard lamellipodia protrusion under high tubular confinement*".

Futhermore, the fact that tissue forward polarization was poorer in smaller tubes was in line with the measurement of taller cell height (new manuscript Fig. 2c) and lower tissue speeds in smaller microtubes (new manuscript Fig. 4d, H1-GFP). A fused, multilayer structure frequently seen in smaller microtubes (new manuscript Fig. 2a and Supplementary Fig. 4) could also explain the less pronounced front-back polarization there.

Fig. R1: Taken from new manuscript Fig. 8. (a) Representative images of PBD-YFP cells in 75 and 150 μm diameter tubular tissues moving upward (blue arrow). Green and cyan (red) arrows show the active PBD zones as marker of lamellipodia protrusion in (opposite to) the direction of tissue front (yellow dashed line) expansion. Time given in hr:min, scale bars: 40 μm . (b) Calculation of the average lamellipodia persistence parameter, $v_p = \sum_i v_i / \sum_i |v_i|$, as function of tube diameter, where v_i are all the velocity vectors falling into the zones marked by the active PBD signal (see methods for how the zones are determined). Each point denotes an experiment.

1a. Also, why does mitomycin C have such a strong effect on cell speed? Does cell proliferation increase in cylindrical microtubes?

Response:

Fig. R2: Taken from new manuscript Fig. 6d, b, and f, as well as Supplementary Fig. 11. (a) Average velocity in fixed region related to division as function of time, further averaged over many division events. (b) Tubular confinement does not affect MDCK proliferation. (c) Average Distance Perturbation Factor as function of tube diameter. (d) Speed-ratio = \bar{v}_f (without proliferation) / \bar{v}_f (with proliferation) for WT MDCKs and H1-GFP MDCKs in various conditions. t -test, $*P < 0.05$, $***P < 0.001$, NS, non-significant. All data are presented as mean \pm s.e.m.

As the reviewer has pointed out, mitomycin C treatment increases the average front velocity (\overline{v}_f) in all conditions. After carefully analyzing the velocity field of TCSs, we found that cell division events perturb and slow down local velocity fields pointing in the direction of tissue expansion for all microtube sizes (Fig. R2a). This is consistent with previous simulation studies showing that cell divisions can fluidize and increase tissue viscosity^{1,2}. Thus, mytomycin C treatment could act to reduce tissue viscosity generated by cell division, which can dissipate the active tissue propulsion energy and tissue speed². This would then lead to overall speed increase. In addition, we demonstrated that the percentage of proliferating cells (over total number of cells) were similar for microtubes of all sizes using Edu staining³ (Fig. R2b). However, the average division-induced velocity perturbation was found to be higher in smaller microtubes (quantified by Distance Perturbation Factor, Fig. R2c), which can explain why mitomycin C treatment has a stronger effect in these tubes (Fig. R2d). Detailed arguments are now presented in the new manuscript in pages 11 – 12 and in the section of “Cell division’s influence on local velocity field”.

1b. How does myosin activity change as the tube diameter increases?

Response:

We controlled myosin activity in the TCSs by systematically varying the blebbistatin concentration up to 100 μM , and studied the drug effect using tissue expansion speed as a read-out in microtube of all sizes. The results demonstrated that tissue speeds in all microtubes increased in general with increasing concentration up to 50 μM blebbistatin (Fig. R3). However, cell migration in the smallest tube (25 μm) presents a non-monotonic variation with increased blebbistatin, as opposed to larger tubes (above 50 μm) in which the velocity monotonically increases as a function of blebbistatin. The speed trend can be separated into three distinct groups based on microtube size, for diameter = 25 μm , diameter \leq 50 - 100 μm , and diameter > 100 μm (Fig. R3), suggesting that there are indeed differences in contractility (related tissue parameters) between smaller and bigger microtubes.

Fig. R3: Taken from new manuscript Fig. 9. Tissue expansion speed as a function of blebbistatin concentration can be separated into three groups. (a) 25 μm ($n = 7$ from 3 independent experiments in each condition), (b) 50 – 100 μm ($n = 6$ from 3 independent experiments in each condition), and (c) 150 and 250 μm ($n = 6$ from 3 independent experiments in each condition). Data are presented as mean \pm s.e.m.

The trends of the graph could be related to changes in global tissue elasticity, E and active tissue propulsion stress, σ_p at the tissue front which can both be reduced by decreasing acto-myosin contractility⁴. Specifically, the increasing velocity in the large tubes may be related to a reduced tissue elasticity under blebbistatin that may also favour protrusive activity through actin polymerisation and thus leading front migration. For narrow channels, the blebbistatin treatment may not be sufficient to promote enough driving force and cell protrusions at the monolayer front to overcome resistance from tissue elasticity. This is consistent with the results that cells polarized less in smaller tubes (Fig. R1) and spread poorly on bowl-shaped structures with high negative curvature in all directions (new manuscript supplementary Fig. 6c, 50 μm diameter bowl shape). Compared to the bowl structure, microtubes only have negative curvature in the circumferential direction (but zero curvature on the longitudinal direction), but since cells could spread less in one direction (circumferential), this could induce taller cell heights in the smallest microtubes with the highest negative curvature.

Overall, the analysis of blebbistatin drug treatment results suggested that tissue elasticity and active tissue propulsion governed by actomyosin contraction and actin polymerisation are both at lower levels in the smallest microtubes than in larger ones. Also, since the contractility behaviours can be distinguished into three groups based on microtube size (Fig. R3), the middle-sized microtubes should have an intermediate contractility behavior (diameters between 50 – 100 μm) between the smallest and biggest microtubes. Detailed arguments are presented in the new manuscript in pages 14 – 15 and in the section of “Acto-myosin contractility difference leads to variations in TCS average front speed”.

1c. How does the loss of α -catenin induce loss of actin alignment?

Response:

The formation of thick, long and aligned actin fibers usually require the fibers to connect from one cell to the other through mature cell-cell junctions, as can be seen for WT MDCK cells in microtubes (new manuscript Fig. 3a and supplementary Fig. 7a) and other epithelial structures such as epithelial bridges hanging over non-adherent surfaces⁵. As α -catenin is a tension transducer⁶ that can strengthen cell-cell junctions by recruiting more actin fibers (under the α -catenin open conformation and interaction with Vinculin⁷) at the junctional location, its knockdown will destabilize actin connection between neighboring cells at the cell-cell junction, and could thus induce loss of actin alignment.

Since actin orientation usually follow the cell body alignment⁸, the fact that cell orientation is randomized in α -catenin knock down tubular tissues (new manuscript Fig. 7c) could also contribute to less actin alignment.

1d. Why does loss of α -catenin decrease migration speed in larger diameter tubes and not in smaller tubes?

Response:

The reason for this could be that the α -catenin knock down still impacts on collective migration in smaller tubes by weakening the tissue elasticity, E , but its effects were less prominent for tissue migration speed results (new manuscript Fig. 4d) as the H1-GFP MDCK migration speed was already very low in the smallest microtubes. To confirm this, we redid tissue migration experiments in microtubes of all sizes using H1-GFP and WT-MDCK (Fig. R4a) and found that the tissue speeds in all microtubes were similar to those for H1-GFP MDCK.

To confirm that the loss of α -catenin still has impact on the smallest microtubes, we further checked that the velocity spatial correlation length for α -catenin knock down is consistently lower (in microtube sizes of 50, 100 and 150 μm diameters) than their WT counterparts in the same microtube sizes (Fig. R4b).

Detailed description of data are presented in the new manuscript in pages 12 – 14 and in the section of “Role of cell-cell junction in collective TCS migration dynamics”.

Fig. R4: Taken from new manuscript Supplementary Fig. 9 and Fig. 7f. (a) Average velocity of tissue front for WT-MDCKs (black) in tubes of different diameters. For each condition, t -test between each microtube diameter and 25 μm , unless otherwise indicated by lines, $**P < 0.01$, $***P < 0.001$, NS non-significant. The plots

represent the mean \pm s.e.m. (b) Longitudinal velocity correlation length for α -catenin knock down and WT tissue in phase contrast movies of microtubes of diameters 50, 100, and 150 μm .

1e. Does loss of α -catenin “resemble” Snail overexpression? Although the authors have performed experiments with Snail overexpressing cells, they do not provide any quantitative analysis; at least, speed measurements are necessary.

Response:

As shown in our Supplementary Movie 8, Snail overexpression induces the MDCK cells to become more mesenchymal-like and undergo mainly single-cell migration. Thus, Snail overexpression is more severe in disrupting the collective migration of MDCK tissues than loss of α -catenin, which still portrayed certain levels of collective behavior.

As requested by the reviewer, we quantified the dynamics of the Snail-overexpressed cells by tracking the individual cells' tracks and calculating the instantaneous velocities and speeds along these tracks. From a plot of the instantaneous speeds (Fig. R5a), we see that the single cell speeds increased with increasing microtube diameters, and reached a saturation point at diameter $\sim 75 - 100 \mu\text{m}$. Interestingly, this mirrored the trend found for tissue expansion speeds as function of microtube diameter for MDCKs (new manuscript Fig. 4d and Supplementary Fig. 9). Since Snail-overexpressed cells behave more as single cells, this suggested that the observed tissue expansion speed trend could be partially understood through parameters found also in single cells such as poorer lamellipodia protrusion and cell polarization (see PBD results and Fig. R1), and the possibility of lower active cell propulsion stress (interpretation for blebbistatin analysis) in smaller microtubes.

Further, from a plot of the Snail instantaneous velocity along the microtube long axis (Fig. R5b), it was found that these velocities distributed mostly symmetrically about the zero value for all microtube sizes, which would lead to no average net movement anywhere, unlike other tissues which persistently migrated forward.

Detailed argument is presented in the new manuscript in pages 12 – 14 and in the section of “*Role of cell-cell junction in collective TCS migration dynamics*”.

Fig. R5: Taken from new manuscript Fig. 7g and h. (a) Bar graphs showing average of instantaneous speeds along Snail-overexpressed cell tracks, for different microtube diameters, (25 μm : $n = 11$, 50 μm : $n = 12$, 75 μm : $n = 16$, 100 μm : $n = 16$, 150 μm : $n = 16$, 250 μm : $n = 16$ cell tracks of varying duration between 100 – 800 min, from 2 independent experiments per condition). Data presented as mean \pm s.e.m. (b) Scatter plot showing the longitudinal velocities along the Snail-overexpressed cell tracks.

2. Statistical analysis is missing throughout the manuscript.

Response:

We have added statistical analysis (t -test and statistical significance) in all relevant data.

3. The authors need to provide an explanation on why the height of the cells increase as the tube diameter decrease.

Response:

As explained above, tubes have negative curvature in the circumferential direction (but zero curvature in the longitudinal direction), and as tube diameter decreases, the negative curvature increases in magnitude. Since cells are known to spread less on bowl-shaped structures with negative curvature in all directions^{9,10} as also observed in MDCK cells in our additional experiments using 50 μm diameter bowl structures (new manuscript Supplementary Fig. 6c), this could explain the taller cell height observed in smaller microtubes.

There is also possibility that a significant radial force exists pointing toward the center axis of the microtube that can stretch the cell height (Fig. R6). This radial force on each cell can originate from tensile stress from neighboring cells in the circumferential direction leading to a resultant upward stress, which is proportional to curvature. Such stress in the TCSs may explain why cell height is taller in smaller microtubes with higher curvature.

A similar argument is presented in the new manuscript in page 6 and in the section of “*Negative curvature and tubular confinement influence epithelial TCS organization*”.

Fig. R6: Taken from new manuscript Supplementary Fig. 6d. Schematic cross-section view of tissue in tube, where the resultant (red, pointing toward the midpoint of the tube) force on a cell arises from the tensile stress in the circumferential direction (black) from both neighbors.

4. Are the authors' observations cell line specific? Experiments with additional cell lines are required to generalize findings.

Response:

We thank the reviewer for this constructive suggestions. Our observations are not cell line specific, as we did experiments with WT-MDCK (Fig. R4a) and MCF10A (Fig. R7), which both showed tubular tissue formation and expansion speeds that decreased with increasing microtube confinement.

Fig. R7: Taken from new manuscript Supplementary Fig. 10a and b. (a) Average velocity of tissue front for MCF-10A cells in tubes of different diameters. (b) 3D fluorescent reconstructed image of MCF-10A forming proper tubular structures in 100 μm diameter microtube. DAPI – blue, Phalloidin - green, scale bar: 50 μm. *t*-test, * *P* < 0.05, ** *P* < 0.01, NS non-significant. The plots represent the mean ± s.e.m.

5. Since the authors use PDMS (which is very stiff) to study collective migration, do they expect similar results under physiologically relevant stiffnesses? How does substrate stiffness affect collective migration in microtubes?

Response:

We note that while the study of changing mechanism in collective migration due to various stiffness is certainly important, the study focuses on the yet unknown effects of the unique surface curvature found in tubular structures on tissue expansion dynamics. Thus, we used stiff PDMS substrates where cells were sure to spread properly in the flat surface case, which allowed us to compare the results between curved and flat surfaces *in vitro*. Using a soft substrate to fabricate the microtubes is not only highly challenging technically, but would also constitute a confounding factor for our interpretation of the current results. This is because substrate stiffness effects on cell spreading/migration on flat surfaces are still unclear, and only few studies have tried to study this systematically to date^{11,12}. Even so, these studies were done only for single cell types, and collective cell migration is arguably more difficult to dissect due to additional cell-cell interactions on top of cell-substrate interactions. Therefore, it would be interesting to explore the effects of substrate stiffness on collective tubular tissue migration in the future especially due to its physiological relevance, but it is out of scope for the current study.

6. The authors use the same control cell line for the inhibitor, the knockdown and overexpression studies. The appropriate control cell line needs to be used for each experiment.

Response:

We agree with the reviewer that the previous version lacked proper control cell line in our experiment. However, we have used WT-MDCK as the appropriate control cell line and we see similar tissue dynamics results (Fig. R4a) with the H1-GFP MDCK cells. The new data have been included in the revised manuscript.

Minor comments:

1. Page 17 line 493, $\text{Cos}(2\phi)$ needs to be replaced with $\text{cos}(\phi)$.

Response:

We have corrected this typo.

2. Page 17 line 504, a description of C_u and C_v is missing.

Response:

It is given in the paragraph below the mathematical definition of the correlation function.

3. Correlation length needs to be better explained.

Response:

We have updated the main text in page 22 and in the section of “*Methods*” to clearly explain how the velocity correlation length was determined:

“If two vectors are correlated, one of the vector values and directions can be predicted by knowing the other vector. Two vectors which are closer spatially to each other usually have higher correlation with one another, compared to two vectors further apart. The average correlation magnitude between two vectors that are separated by a distance $|\vec{r}|$ is given by the correlation function defined as¹²:

$$C_u(\vec{r}) = \left\langle \frac{\langle u^*(\vec{r}' + \vec{r}, t) \times u^*(\vec{r}', t) \rangle \vec{r}'}{[\langle u^*(\vec{r}' + \vec{r}, t)^2 \rangle \langle u^*(\vec{r}', t)^2 \rangle]^{1/2}} \right\rangle t$$

$$C_v(\vec{r}) = \left\langle \frac{\langle v^*(\vec{r}' + \vec{r}, t) \times v^*(\vec{r}', t) \rangle \vec{r}'}{[\langle v^*(\vec{r}' + \vec{r}, t)^2 \rangle \langle v^*(\vec{r}', t)^2 \rangle]^{1/2}} \right\rangle t$$

Where C_u and C_v correspond to the velocity correlation functions in the directions that are vertical and parallel to the microtubule longitudinal axis, respectively, u^* and v^* are the deviation of the velocity from the mean velocity in the directions vertical and parallel to the microtubular long axis, \vec{r} is the vector of the coordinates, and t is time. The spatial velocity correlation functions were computed by calculating the mean of the correlation coefficient over all the directions such that C_u and C_v are now functions of $\|\vec{r}\|$ (the norm of \vec{r}). The correlation length then refers to the distance ($\|\vec{r}\|$) where the spatial velocity correlation function first reaches zero.”

4. Page 6 line 150 “is was then”- Please correct this.

Response:

We have made correction to it.

Reviewer #2 (Remarks to the Author):

Up to now, few studies have looked at dynamical aspects of coordinated epithelial behaviors across space and time in non-planar geometries. To address these challenges, this manuscript uses a microtube platform for the study of collective epithelial dynamics that leads to the formation of hollow tubules. The results are novel, interesting, and important. The manuscript suffers from a few typos and grammatical errors, but otherwise is very well-written. I have only a few comments that will need to be addressed.

Response:

We thank the reviewer for his/her positive feedback. We address below several comments by the reviewer.

Lines 71-74. These arguments fail to thread the needle. Yes, gel and matrix based 3D assays are more complex, and the geometry is self-assembled, but these situations are closer to the real biology. In your preparation you gain control of the curvature but move away from the biology. You should probably just say that.

Response:

We thank the reviewer for pointing this out. After carefully consideration and to avoid confusion, we have removed the comment in the main text.

Line 220: “As shown in Fig. 4d, this increased the average front edge velocity in all conditions.” This is counter-intuitive, no? Is it due to an off-target effect. Some comment is warranted here is very much needed.

Response:

The reviewer is questioning whether mitomycin C treatment (cell division inhibition) increasing cell speed in all tubes (new manuscript Fig. 4d) is due to off-target effects, similar to **Reviewer 1's question 1a**. We speculate that the confusion regarding this division inhibition result might come from the understanding that cell division serves to add mass to the migrating tissue, and should help the tissue attend a larger expansion length, as also quantified in other simulation and experimental studies^{2,13}. However, tissue speed is the rate of tissue expansion, and is different from the extent tissue expansion (total tissue length). Our tubular tissue is also connected to a tissue reservoir (new manuscript Fig. 1a) that helps to contribute the cell mass needed for tubular tissue expansion.

In Fig. R2a, we showed that cell division events perturb and slow down local velocity fields pointing in the direction of tissue expansion for all microtube sizes. Since mitomycin C inhibits cell divisions, the drug would reduce perturbation to cell migration and thus increase tissue speed. Simulation studies also showed that cell divisions can fluidize and increase tissue viscosity^{1,2}, mytomycin C treatment could also act to reduce tissue viscosity generated by cell division, which can dissipate the active tissue propulsion energy and tissue speed². This would then lead to overall speed increase.

In Fig. R2b, we showed that the percentage of proliferating cells (over total number of cells) were similar for microtubes of all sizes using EdU staining³. However, the average division-induced velocity perturbation was found to be higher in smaller microtubes (quantified by Distance Perturbation Factor, Fig. R2c), which can explain why cell speeds were slower in these tubes. Detailed arguments are presented in the new manuscript in pages 11 – 12 and in the section of “*Cell division's influence on local velocity field*”.

Line 228 “The inhibition of myosin-II through blebbistatin treatment increased the velocity in all tubes” This is counter-intuitive as well. Some comment is warranted.

Response:

As tissue speed can vary in complex ways with contractility^{2,14,15}, we did additional experiments by treating tissues in all microtube sizes with varying blebbistatin concentration up to 100 μ M, and studied the tissue expansion speed. Focusing on the global trend, tissue speeds in all microtubes were found to increase in general with

increasing blebbistatin concentration up to 50 μM blebbistatin (Fig. R3). However, while the tissue speed for 25 μm microtube reduced back to its value for no drug treatment at 100 μM blebbistatin, tissue speeds in larger microtubes continued to increase. The trends of the graph could be related to changes in global tissue elasticity, E and active tissue propulsion stress, σ_p at the tissue front which can both be reduced by decreasing acto-myosin contractility⁴. Specifically, the increasing velocity in the large tubes may be related to a reduced tissue elasticity under blebbistatin that may also favour protrusive activity through actin polymerisation and thus leading front migration. For narrow channels, the blebbistatin treatment may not be sufficient to promote enough driving force and cell protrusions at the monolayer front to overcome resistance from tissue elasticity. This is consistent with the results that cells polarized less in smaller tubes (Fig. R1) and spread poorly on bowl-shaped structures with high negative curvature in all directions (new manuscript Supplementary Fig. 6c, 50 μm diameter bowl shape). Compared to the bowl structure, microtubes only have negative curvature in the circumferential direction (but zero curvature on the longitudinal direction), but since cells could spread less in one direction (circumferential), this could induce taller cell heights in the smallest microtubes with the highest negative curvature.

Overall, the analysis of blebbistatin drug treatment results suggested that tissue elasticity and active tissue propulsion governed by actomyosin contraction and actin polymerisation are both at lower levels in the smallest microtubes than in larger ones. Also, since the contractility behaviors can be distinguished into three groups based on microtube size (Fig. R3), the middle-sized microtubes should have an intermediate contractility behaviour (diameters between 50 – 100 μm) between the smallest and biggest microtubes. Detailed arguments are presented in the new manuscript in pages 14 – 15 and in the section of “*Acto-myosin contractility difference leads to variations in TCS average front speed*”.

Discussion: When you talk about swirling motions in space and time, I was surprised that you did not mention at all cell jamming effects and associated glassy dynamics that have been well described in the recent literature for flat culture. If you think that the dynamics here are unrelated to those phenomena, you should say so, and defend your assertion. And if you think that these effects are in play, you should say that.

Response:

We thank the reviewer for this suggestion to include cell jamming to explain the slower tissue speeds and swirling motions in smaller microtubes, which we do think is relevant.

We have included these points in the **Results** section (page 8) “The fact that tissue expansion speeds were lower for microtube diameters < 100 μm could be related to cell jamming in these tubes due to higher cell densities and the additional plugging of the smallest microtubes by multi-layered tissue structures⁴⁹⁻⁵¹.”, and in the **Discussion** section (page 16) “Due to this unique structure, cells could be more jammed in the smaller tubular tissues which also consistently portrayed poor forward polarization, marked by haphazard lamellipodia protrusion directions. On flat surfaces, MDCK cells within confluent tissues started to transition from a more directed migration motion to a more jammed, diffusive motion with lower speeds around the cell density threshold of 3 – 4 cells/(1000 μm^2)⁴⁹⁻⁵¹. In contrast, the density threshold that separated smaller microtubes (\leq 100 μm diameter, with lower tissue speeds) from bigger microtubes was measured to be \sim 2 cells/ 1000 μm^2 (Fig. 2b and Supplementary Fig. 6b), which suggested that surface curvature can shift the jamming transition to lower cell densities. The possibility of jamming in smaller tubes also coincided with the observation of backward-forward motion and swirling^{69,70} that predominantly happened in microtube diameters \leq 100 μm (Fig. 5a, d, and e).”

References

1. Ranft J., *et al.* Fluidization of tissues by cell division and apoptosis. *Proc. Natl. Acad. Sci. USA* **107**, 20863-20868 (2010).
2. Recho P., Ranft J., Marcq P. One-dimensional collective migration of a proliferating cell monolayer. *Soft Matter* **12**, 2381-2391 (2016).

3. Chaudhuri P. K., Pan C. Q., Low B. C., Lim C. T. Topography induces differential sensitivity on cancer cell proliferation via Rho-ROCK-Myosin contractility. *Sci. Rep.* **6**, 19672 (2016).
4. Balland M., Richert A., Gallet F. The dissipative contribution of myosin II in the cytoskeleton dynamics of myoblasts. *Eur. Biophys. J.* **34**, 255-261 (2005).
5. Vedula S. R. K., *et al.* Epithelial bridges maintain tissue integrity during collective cell migration. *Nat. Mater.* **13**, 87-96 (2014).
6. Yonemura S., Wada Y., Watanabe T., Nagafuchi A., Shibata M. α -catenin as a tension transducer that induces adherens junction development. *Nat. Cell Biol.* **12**, (2010).
7. Yao J., *et al.* Optical transmission enhancement through chemically tuned two-dimensional bismuth chalcogenide nanoplates. *Nat. Commun.* **5**, 5670 (2014).
8. Gupta M., *et al.* Adaptive rheology and ordering of cell cytoskeleton govern matrix rigidity sensing. *Nat. Commun.* **6**, 7525 (2015).
9. Park J. Y., Lee D. H., Lee E. J., Lee S.-H. Study of cellular behaviors on concave and convex microstructures fabricated from elastic PDMS membranes. *Lab Chip* **9**, 2043-2049 (2009).
10. Werner M., *et al.* Surface curvature differentially regulates stem cell migration and differentiation via altered attachment morphology and nuclear deformation. *Adv. Sci.* **4**, 1600347 (2017).
11. Elosegui-Artola A., *et al.* Rigidity sensing and adaptation through regulation of integrin types. *Nat. Mater.* **13**, 631-637 (2014).
12. Bangasser B. L., *et al.* Shifting the optimal stiffness for cell migration. *Nat. Commun.* **8**, 15313 (2017).
13. Streichan S. J., Hoerner C. R., Schneidt T., Holzer D., Hufnagel L. Spatial constraints control cell proliferation in tissues. *Proc. Natl. Acad. Sci. USA* **111**, 5586-5591 (2014).
14. Vedula S. R. K., *et al.* Emerging modes of collective cell migration induced by geometrical constraints. *Proc. Natl. Acad. Sci. USA* **109**, 12974-12979 (2012).
15. Yevick H. G., Duclos G., Bonnet I., Silberzan P. Architecture and migration of an epithelium on a cylindrical wire. *Proc. Natl. Acad. Sci. USA* **112**, 5944-5949 (2015).

REVIEWERS' COMMENTS:

Reviewer #1 (Remarks to the Author):

The revised manuscript by Xi et al. has been significantly improved. Yet some additional issues need to be addressed before this manuscript reaches publication quality.

Major Comment:

The experiments aimed to elucidate the role of contractility in collective migration in microtubes are incomplete. The authors postulate that in confined microtubes contractility levels are low, and they increase with increasing the tube size. To "prove" their point, the authors performed experiments with different doses of blebbistatin. However, this is an indirect way of "proving" their point. Staining for pMLC is more appropriate. If the contractility levels are indeed low in confined microtubes, then inhibition of myosin activity via the use of blebbistatin should have a minimal effect. To the contrary, the authors report an initial increase and a subsequent decrease in the speed as a function of varying degrees of myosin inhibition.

Minor Comments:

1. Legend of figure 6 (the labelling is wrong)
2. Legend of figure 7 (there is no mitomycin C treatment in this figure)
3. The y axis labels of figure 7 g and h do not accurately describe the plotted variable
4. Is there a significant difference between cnt and mito c treated cells during migration in 25 μ m tubes (Figure 6 a)?
5. What concentrations of blebbistatin did the author use? This should be included in the materials and methods.

Reviewer #2 (Remarks to the Author):

My previous concerns and queries have been suitably addressed.

Point-by-point response to the reviewers' comments

First of all, we would like to thank the reviewers for their careful reviewing of the revised manuscript and their constructive comments, which have allowed us to further revise and improve the manuscript.

Reviewer #1 (Remarks to the Author):

The revised manuscript by Xi et al. has been significantly improved. Yet some additional issues need to be addressed before this manuscript reaches publication quality.

Response:

We thank the reviewer for his/her positive comments. In the latest revised version of the manuscript, we have incorporated new experiments and discussion which provide a better explanation for the role of contractility in collective migration in microtubes.

Major comments:

The experiments aimed to elucidate the role of contractility in collective migration in microtubes are incomplete. The authors postulate that in confined microtubes contractility levels are low, and they increase with increasing the tube size. To “prove” their point, the authors performed experiments with different doses of blebbistatin. However, this is an indirect way of “proving” their point. Staining for pMLC is more appropriate. If the contractility levels are indeed low in confined microtubes, then inhibition of myosin activity via the use of blebbistatin should have a minimal effect. To the contrary, the authors report an initial increase and a subsequent decrease in the speed as a function of varying degrees of myosin inhibition.

Response:

As suggested by the reviewer, we have performed a careful investigation into the pMLC distribution in MDCK cells that collectively migrated into various microtubes by staining for pMLC and comparing the fluorescent intensity in luminal structure with that on the flat condition. The new data are now presented in Fig. 9a and b. Our results demonstrate that pMLC signal is diffuse, non-colocalizing with actin stress fibers and weak in the intensity while it is strong and colocalized with actin stress fibers in larger tubes, similar to those on flat substrates. By calculating the relative fluorescent intensity (I_r), we found an increasing trend from low I_r in the smallest microtube to high and plateaued I_r in larger microtubes. This indicates that the acto-myosin contractility level is indeed low in highly confined (smaller) microtubes.

Based on these additional understanding of acto-myosin contractility, we have also rewritten the part on the blebbistatin treatment (now Fig. 9c – e) to make it clearer. The observation of an increase in tissue expansion speed in all tubes at lower blebbistatin concentrations could be related to decrease in tissue elasticity that allows the tissue to extend more easily^{1,2}. Importantly, the fact that only tissue speed in the smallest microtubes decreased (and dropped to low values) at high blebbistatin concentrations shows that such tissues have the lowest myosin-II contractile activity levels and are the first to be depleted of activity. This is consistent with the pMLC data. Detailed arguments are now presented in the revised manuscript in the section of “Contractility difference leads to migration speed variations”.

Minor Comments:

1. Legend of figure 6 (the labelling is wrong)

Response:

We have made changes to the legend and the labelling has been corrected.

2. Legend of figure 7 (there is no mitomycin C treatment in this figure)

Response:

We have deleted "...and mitomycin C treated H1-GFP MDCKs..." in the legend to avoid confusion.

3. The y axis labels of figure 7 g and h do not accurately describe the plotted variable

Response:

We have changed the y-axis label for Fig. 7g to "Average Migration Speed ($\mu\text{m}/\text{min}$)", as the plot presents the average speed along cell tracks. Similarly, we changed the y-axis label for Fig. 7h to "Instantaneous Longitudinal Velocity ($\mu\text{m}/\text{min}$)", as the plot denotes the instantaneous single cell velocity in longitudinal direction of the microtubes. We also add label – "Snail MDCK" to Fig. 7h for clarification.

4. Is there a significant difference between cnt and mito c treated cells during migration in 25 μm tubes (Figure 6 a)?

Response:

We have added *t*-test and statistical significance between cnt and mitomycin C treated cells in 25 μm tube in Fig. 6a.

5. What concentrations of blebbistatin did the author use? This should be included in the materials and methods.

Response:

We have added the exact blebbistatin concentrations (6, 12.5, 25, 50, and 100 μM) in the section of "Cell culture and imaging".

Reviewer #2 (Remarks to the Author):

My previous concerns and queries have been suitably addressed.

Response:

We again thank the reviewer for his/her positive feedback. His/her comments have helped to greatly improve our manuscript.

References

1. Schillers H., Wälte M., Urbanova K., Oberleithner H. Real-time monitoring of cell elasticity reveals oscillating myosin activity. *Biophys. J.* **99**, 3639-3646 (2010).
2. Recho P., Ranft J., Marcq P. One-dimensional collective migration of a proliferating cell monolayer. *Soft Matter* **12**, 2381-2391 (2016).